# A library of 2D electronic material inks synthesized by liquid-metal-assisted intercalation of crystal powders

Shengqi Wang[1], Wenjie Li[1], Junying Xue[1], Jifeng Ge[1], Jing He[1], Junyang Hou[1], Yu Xie[1], Yuan Li[1], Hao Zhang[1], Zdeněk Sofer[2] & Zhaoyang Lin[1] ✉

Solution-processable 2D semiconductor inks based on electrochemical molecular intercalation and exfoliation of bulk layered crystals using organic cations has offered an alternative pathway to low-cost fabrication of large-area flexible and wearable electronic devices. However, the growth of large-piece bulk crystals as starting material relies on costly and prolonged high-temperature process, representing a critical roadblock towards practical and large-scale applications. Here we report a general liquid-metal-assisted approach that enables the electrochemical molecular intercalation of low-cost and readily available crystal powders. The resulted solution-processable $MoS_2$ nanosheets are of comparable quality to those exfoliated from bulk crystals. Furthermore, this method can create a rich library of functional 2D electronic inks ( >50 types), including 2D wide-bandgap semiconductors of low electrical conductivity. Lastly, we demonstrated the all-solution-processable integration of 2D semiconductors with 2D conductors and 2D dielectrics for the fabrication of large-area thin-film transistors and memristors at a greatly reduced cost.

Solution-processable two-dimensional (2D) semiconductors have been considered promising for the fabrication of large-area flexible and wearable electronic devices at an affordable cost[1–4]. Compared with the conventional vapor-based chemical vapor deposition (CVD) that involves high-temperature (>500 °C) and/or vacuum process in the furnace, the solution-based approaches offer an alternative pathway to the scalable and cost-effective production of 2D semiconductors over a large area[5–7]. With the easy-to-handle ink materials, the 2D semiconductors may be processed by high-throughput and low-cost solution-based material deposition methods such as spin coating, ink printing, and roll-to-roll fabrication. By circumventing high-temperature and vacuum-based material manufacturing processes, it becomes feasible to produce large-area and flexible 2D semiconductor electronic devices with greatly reduced manufacturing cost[7–12]. Throughout the solution-based fabrication process, the formulation of high-quality solution-processable 2D semiconductor ink materials is the key to producing compact, uniform, and impurity-free thin films[6,8,13]. To this end, it has been attracting significant research interests in developing efficient chemical approaches to high-quality semiconductor ink materials and high-performance large-area electronic devices using low-temperature solution-based approaches.

Currently, the formulation of semiconductor ink materials typically relies on two types of chemical methods, including soluble molecular precursors and colloidal solutions of free-standing 2D nanosheets. The solution of molecular precursors (e.g., $(NH_4)_2MoS_4$ dissolved in dimethylformamide for $MoS_2$) can be used as the ink material for depositing thin films. But it usually requires high annealing temperature (e.g., >500 °C) to decompose these precursors into 2D semiconductor thin films[14,15]. Therefore, this method is not favored for flexible plastic substrates and also for cost-effective material production. On the other hand, the colloidal solution of 2D nanosheets prepared by the top-down liquid-phase exfoliation of bulk layered crystals

[1]Department of Chemistry, Engineering Research Center of Advanced Rare Earth Materials (Ministry of Education), Tsinghua University, Beijing 100084, China. [2]Department of Inorganic Chemistry, University of Chemistry and Technology Prague, Technická 5, 166 28 Prague 6, Czech Republic. ✉e-mail: zlin@mail.tsinghua.edu.cn

has offered a promising route to the solution-processable ink materials with low processing temperature (e.g., <350 °C) and compatibility with flexible plastic substrate[7,8,11,16–18]. For improved synthetic control of the thickness and material uniformity, the chemical or electrochemical intercalation of inorganic alkali metal ions (e.g., $Li^+$, $Na^+$) into layered bulk crystals for subsequent liquid-phase exfoliation has been a prevailing method[3,19–23]. Recently, to minimize the structural alteration caused by violent alkali metal ion intercalation and gas production process, the electrochemical intercalation of bulky organic cations (e.g., quaternary alkylammonium cations) emerged as an alternative pathway to high-quality 2D semiconductor nanosheets[2,24–30]. Following this method, 2D semiconductors such as $MoS_2$, $WSe_2$, and black phosphorus have been intercalated and exfoliated into high-quality solution-processable 2D inks. The fabricated transistors based on low-temperature-processed thin films (~300 °C) have shown promising electrical performance[2,27,31,32].

Despite the great potential in producing high-quality solution-processable 2D semiconductors, the electrochemical molecular intercalation and exfoliation process usually uses large-size bulk single crystals (e.g., crystal length >3 mm) as the source material[25,33]. However, the growth of these large crystals typically involves high-temperature reactions (>800–1000 °C) and/or prolonged growth processes (>100 h) such as chemical vapor transport and/or melt-based methods, which require sophisticated chemical control and are also expensive for scaled-up and practical material synthesis. Therefore, replacing the expensive large-size bulk single crystals with low-cost and readily available crystal powders (e.g., crystal length <100 μm) has been considered a feasible solution. In 2022, Sivula et al. initially reported an approach to directly hot press fine microcrystal powders into bulk pellets (length ~12 mm) followed by high-temperature annealing at 1100 °C to enhance the powder interaction and mechanical strength[34]. Then the pressed pellet can be intercalated and exfoliated just like the regular large-size single crystals. Despite the high annealing temperature of the pellet, the produced 2D nanosheets, such as $MoS_2$ and $WSe_2$, exhibit lower thickness uniformity and crystal quality than previous materials resulting from bulk crystals and thus are not favored for high-performance and large-area thin-film transistors. Another attempt was made by mixing 2D crystal powders with polymer binders such as poly(vinylidene fluoride) into a slurry which is then coated onto conductive copper foil as the electrode for electrochemical intercalation[24]. However, the insulating polymer binders and the potential organic residues may be problematic for realizing high electrical performance in the resulting films. To this end, a reliable chemical protocol for conducting electrochemical molecular intercalation and exfoliation with low-cost crystal powders is desired for producing 2D semiconductor inks of high material quality and device performance.

Here, we report the robust electrochemical molecular intercalation of 2D crystal powders enabled by the conductive and adhesive liquid metal. The soft and liquid-like framework can accommodate the dramatic volume expansion of layered crystals during the molecular intercalation, while simultaneously maintaining the intimate electrical contact to the microcrystals. Also, the exfoliated 2D nanosheets exhibit a clean crystal structure that is free of liquid metal residues. The exfoliated $MoS_2$ monolayers can be assembled into thin-film transistors with device performance comparable to the previous counterparts resulting from large-size single crystals, including carrier mobility of 10 $cm^2·V^{-1}·s^{-1}$ and a current on/off ratio of $10^6$. Using low-cost and readily available powder materials instead of bulk single crystals greatly expands the scope and applicability of the electrochemical molecular intercalation and exfoliation method. For example, a library of 2D electronic material inks containing >50 types of versatile 2D layered crystals have been demonstrated, spanning from transition metal dichalcogenides (TMDs, e.g., $ZrS_2$, $NbSe_2$, and $MoTe_2$), main group metal chalcogenides (e.g., InSe, $SnSe_2$, and $Bi_2Se_3$), ternary layered crystals (e.g., $MnPS_3$ and $ZnIn_2S_4$), layered oxides (e.g., $MoO_3$ and $V_2O_5$), to elemental crystals

(e.g., graphene and phosphorene). Furthermore, we have realized the intercalation and exfoliation of 2D wide-bandgap semiconductors monolayer inks, including GaS, GaSe, $MoO_3$, and others, which were previously challenging in the form of large-size single crystals. At last, with the toolkit of 2D inks, we have further demonstrated the solution-processable integration of 2D semiconductors with 2D conductors and 2D dielectrics for the fabrication of thin-film transistors and memory devices. Together, our study offers a general and high-throughput exfoliation method using low-cost powder materials and establishes a rich library of high-quality 2D electronic material inks for diverse high-performance large-area electronic devices at a greatly reduced cost.

## Results

### Electrochemical molecular intercalation of crystal powders
Our key idea is to use the liquid metal to construct an adhesive and conductive framework for 2D crystal powders that can function like the regular bulk single crystal for electrochemical molecular intercalation (Fig. 1a–d). The used liquid metal here is Galinstan which consists of 68.5% gallium, 21.5% indium, and 10.0% tin by weight. With a low melting point of −19 °C, it is a conductive metal that can flow like liquid at room temperature. In contrast to conventional liquid metals such as mercury, Galinstan is of much lower toxicity and safer to handle. It is also more chemically reactive than mercury and has a native oxide layer on the surface upon exposure to the ambient air[35,36]. This surface oxide layer is critical for rendering the capability to disperse inorganic 2D crystal powders in Galinstan, which itself is not expected to interact strongly with these powders. But when stirring in the air, the fresh gallium in Galinstan is rapidly oxidized to ultrathin self-limiting oxides ($Ga_2O_3$, ~1 nm). Unlike the liquid metal, the sticky oxide layer can cover the inorganic crystal powders[37] and eventually enables the dispersion in the liquid-state Galinstan as a mixture slurry (Fig. 1a, b). As a comparison, when mixed in the air-free glovebox environment to prevent the oxide formation, the powders stay on the surface of liquid metal and do not dissolve into a uniform slurry (Supplementary Fig. 1). A previous work also reported the incorporation of various nonmetallic materials such as graphite and graphene oxide into Gallium-based liquid metal and discussed the importance of surface oxide in the formation of putty-like composites[38]. The mixing process typically takes a few minutes at room temperature, which is much more convenient and cost-effective than the previous hot-pressing method involving thermal annealing at 1100 °C[34]. The resulting mixture slurry is much more viscous than the pristine Galinstan and thus can be coated onto various substrates, including silicon, gold, copper, glass, and plastic (Supplementary Fig. 2 and Fig. 1e). For the following study, we will use the gold-plated silicon substrate, as gold and gallium can form $AuGa_2$ intermetallic compound to provide strong adhesion. Then, the mixture slurry of 2D crystal powders and Galinstan coated on the conductive gold-plated substrate is used as the working electrode, resembling the free-standing large single-crystal electrode employed in the standard protocol[2,25]. The mixture slurry contains a large number of powders uniformly dispersed in a liquid metal framework which serves as a material reservoir for the intercalation reaction.

During the electrochemical intercalation process, the negative potential (e.g., −6 V) applied to the liquid metal can electrochemically reduce the surface $Ga_2O_3$ to metallic Ga[39,40]. Then the increased surface tension of Galinstan gradually pushes out the dispersed crystal powders inside the mixture slurry to the metal/electrolyte surface. This process is also confirmed by the macroscopic shrinkage of liquid metal from a spread film to a spherical droplet under applied voltage (Supplementary Fig. 3). The color of liquid metal mixture gradually changes from the original silver to brown (Fig. 1f), signifying the exposure of buried internal powders to electrolyte and the intercalation of alkyl-lammonium cations. The unique advantage of the soft liquid metal is that it can accommodate the dramatic volume expansion during the intercalation of organic alkylammonium cations into crystals while

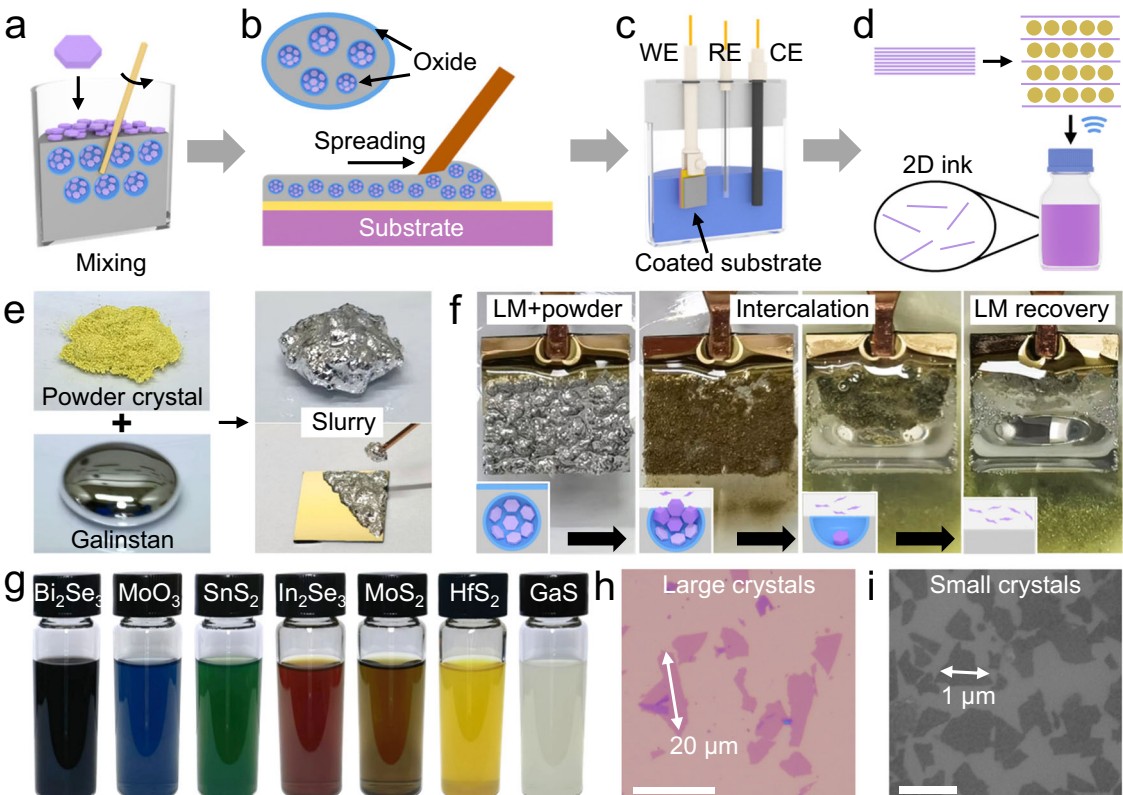

**Fig. 1 | Liquid-metal-assisted electrochemical intercalation and exfoliation of 2D crystal powders. a–d** Schematic illustration of the liquid-metal-assisted electrochemical molecular intercalation of organic cations in 2D layered crystal powders. The liquid metal (Galinstan) was thoroughly mixed with 2D crystal powders into a homogeneous slurry (**a**). The slurry is then coated onto a conductive gold-plated silicon substrate (**b**) as the working electrode for the electrochemical molecular intercalation of tetraheptylammonium cations (THA$^+$) (**c**) and exfoliation (**d**). WE working electrode, RE reference electrode, CE counter electrode. **e** Photographs of the pristine GaS crystal powders and pure Galinstan as well as the viscous mixed slurry and slurry-coated substrate. The powder pile, liquid metal droplet, and mixture slurry are ~10 mm in diameter. **f** The slurry-coated substrate during the electrochemical molecular intercalation. With the completion of the reaction and consumption of all the powders contained in the slurry, the pure liquid metal can be recovered and then used for the next batch of intercalation reaction. The size of the gold-plated silicon substrate is about 12*12 mm. Insets are schematic illustrations of the intercalation and release processes of crystal powders within the mixed slurry. **g** Photographs of representative 2D ink materials obtained by liquid-metal-assisted electrochemical intercalation and exfoliation of crystal powders. **h, i** Optical (**h**) and scanning electron microscopy (**i**) images of exfoliated 2D MoS$_2$ nanosheets with tunable lateral size of 1–20 μm obtained from source powders of different granule sizes. Scale bars, 20 μm (**h**) and 1 μm (**i**).

simultaneously maintaining intimate electrical contact with these microcrystal powders[41,42]. In a standard bulk pellet formed by pressing powders into an entirety, the intercalation and volume expansion of an individual microcrystal might break the pellet structure, resulting in the quick detachment of neighboring powders and a failed powder intercalation. By contrast, the adhesive framework of liquid metal can hold these microcrystal powders throughout the whole intercalation process. When exposed to the electrolyte, the oxide on top is electrochemically reduced to expose the buried microcrystals for intercalation. In the meantime, the oxide on the bottom is not yet reduced and can still connect the microcrystal to the main body of liquid metal. Once the crystals on the surface are fully intercalated and detached, the underlying fresh crystals are then pushed outwards to the solid/electrolyte interface for a continuous intercalation reaction. As a result, the internal 2D layered crystal powders are step by step intercalated until all the powder feedstocks in the mixture slurry are consumed. Finally, the viscous mixture slurry restores the original pure liquid metal state with a shiny silver-like surface and liquid-like fluidity (Fig. 1f). With a much smaller viscosity than the starting mixture slurry, the pure liquid metal shrinks to a droplet-like shape rather than remaining spread over the substrate. It is also worth noting that molecular intercalation and Galinstan oxide reduction would not occur in the absence of external potential (Supplementary Fig. 4). Importantly, the recovered liquid metal can be collected and then

mixed with fresh powders for the next batch of intercalation (Supplementary Fig. 5). The recyclability of the liquid metal can greatly reduce the material cost of the overall material production process which is desired for practical applications.

After intercalation, the expanded crystals settled down at the vial bottom and can be collected and washed with ethanol before sonication-assisted exfoliation in pure dimethylformamide (DMF) liquid or polyvinylpyrrolidone solution in DMF (PVP/DMF) (Fig. 1f, g). The polymer surfactant PVP is needed for some crystals such as MoS$_2$, HfS$_2$, GaS, and Bi$_2$Se$_3$ to stabilize the exfoliated thin nanosheets, while not for others such as In$_2$Se$_3$, SnS$_2$, and MoO$_3$. After sonicating the intercalated materials for about 5 minutes, a homogeneous colloidal solution of 2D nanosheets dispersed in DMF can be obtained. The powder-based method has offered an additional degree of freedom in tunning the material morphology of the obtained 2D monolayers, which was not achievable in the conventional approach using large-size single crystals. For example, by choosing source powders with various particle sizes/lengths (e.g., 2 μm and 80 μm), the exfoliated 2D nanosheets show an average lateral size of ~1 μm and ~20 μm, respectively (Fig. 1h, i). It suggests that the exfoliated 2D nanosheets may partially inherit the lateral size of the starting crystals, offering the extra capability of tuning the lateral size of exfoliated 2D nanosheets that were previously unattainable. Furthermore, 2D nanosheets with regular geometric shapes can be exfoliated when starting with WS$_2$ and

WSe$_2$ hexagonal microcrystals (Supplementary Fig. 6). It has been previously challenging with most liquid-phase exfoliation methods, including LPE or intercalation-assisted exfoliation[1,19,25]. Typically, with the use of large-size bulk single crystals, the exfoliated 2D nanosheets exhibit irregular geometric shapes, which are mostly determined by the molecular intercalation kinetics and non-directional random lattice breakage caused by ultrasonication waves. By contrast, when using small microcrystals (1–10 μm) as the starting material, the mechanical force-induced breakage is suppressed due to the small lateral size that is comparable to the final product. As a result, the pristine hexagonal shape of starting microcrystals is inherited by the exfoliated nanosheets.

**Quality assessment of MoS$_2$ nanosheets exfoliated from powder**
Compared with the conventional protocol of using large-size single crystals, the powder-based intercalation results in 2D nanosheets of comparable morphology, uniformity, and electrical performance.

Taking MoS$_2$ as a typical example, when starting from powder consisting of microcrystals (particle size ~10 μm) (Fig. 2a), the exfoliated 2D nanosheets exhibit lateral size of ~0.5–2 μm and thickness of ~2.8 nm (Fig. 2b). Similar to the previous observation in MoS$_2$ and In$_2$Se$_3$, the 2.8 nm thickness indicates the monolayer structure (~0.6 nm) capped with organic ammonium molecules and PVP surfactants (~2.2 nm) which will be further discussed later[43,44] (Supplementary Fig. 7). The monolayer purity of the exfoliated MoS$_2$ nanosheets is >98%. These values are comparable to those obtained with conventional large-size single crystals (crystal length ~10 mm) (Fig. 2e, f). The morphology of the monolayer nanosheets is reproducible from batch to batch (Supplementary Fig. 8). Although bulk single crystals have much larger crystal domains, the lateral size of the produced nanosheets after intercalation and exfoliation is mainly determined by the molecular intercalation kinetics and mechanical force induced by ultrasonication. Therefore, the lateral size and thickness of the MoS$_2$ monolayer nanosheets exfoliated from crystal

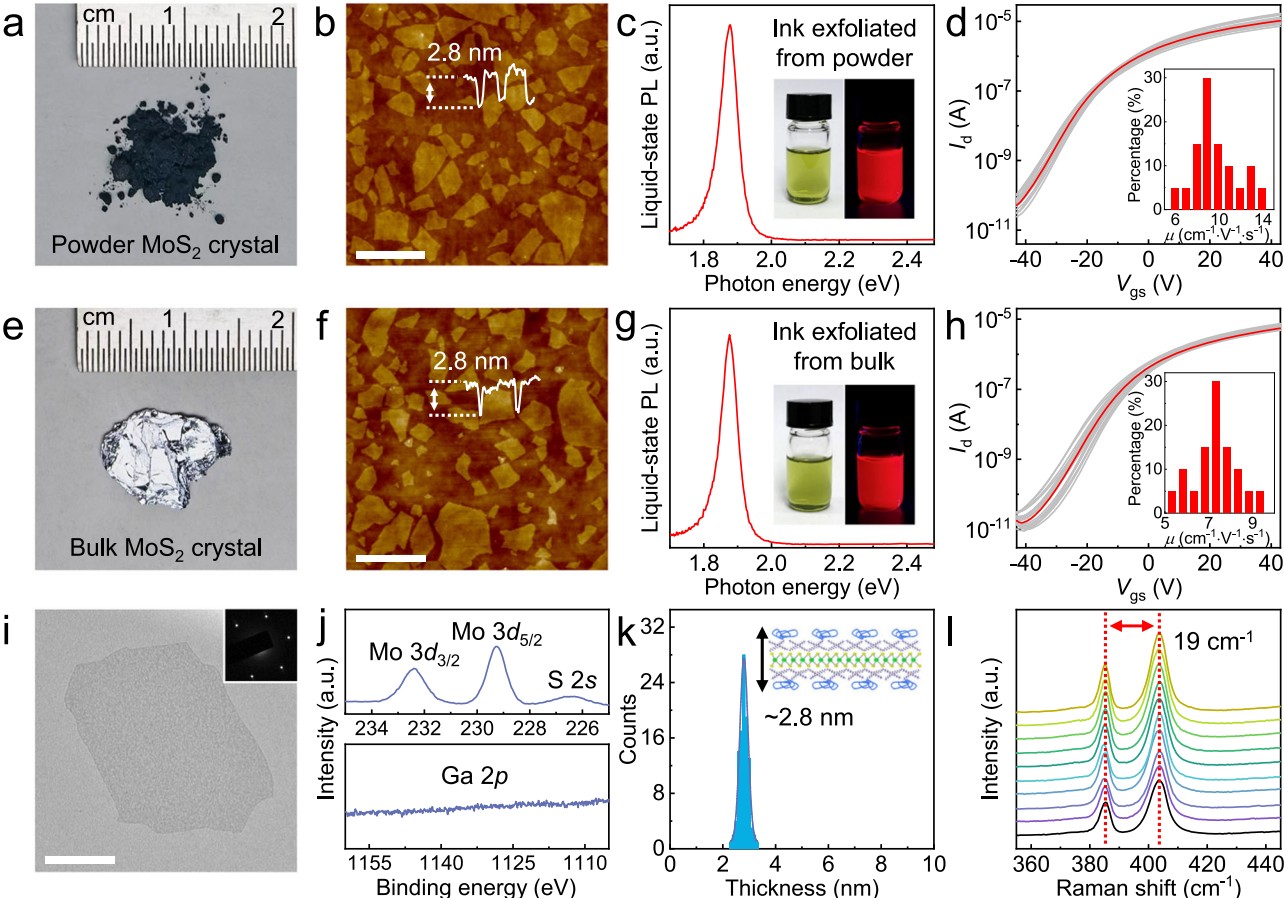

**Fig. 2 | Characterizations of exfoliated MoS$_2$ monolayers from crystal powders.** **a** Photograph of MoS$_2$ crystal powders used for electrochemical molecular intercalation. The average particle size of the powder is ~10 μm. **b** The atomic force microscopy (AFM) image of 2D monolayer nanosheets exfoliated from crystal powders. Scale bar, 2 μm. **c** Liquid-state photoluminescence spectrum of powder-exfoliated MoS$_2$ monolayers dispersed in dichloroethane with bis(tri-fluoromethane)sulfonimide (TFSI) treatment. Insets are the photographs of the ink solution with and without ultraviolet light illumination. The term "a.u." denotes "arbitrary units". PL photoluminescence. **d** $I_d$-$V_{gs}$ transfer characteristics of 20 individual MoS$_2$ transistors based on 2D nanosheets exfoliated from crystal powders. Inset is the statistical distribution of the mobility values for these transistors. $V_{ds}$ = 1 V. $I_d$, drain current; $V_{gs}$, gate-source voltage; $V_{ds}$, drain-source voltage. **e** Photograph of a piece of MoS$_2$ bulk single crystal. **f** The AFM image of 2D monolayer nanosheets exfoliated from the bulk single crystal. Scale bar, 2 μm.

**g** Photoluminescence spectrum of bulk-crystal-exfoliated MoS$_2$ monolayers dispersed in dichloroethane with TFSI treatment. Insets are the photographs of the ink solution with and without ultraviolet light illumination. **h** $I_d$-$V_{gs}$ transfer characteristics of 20 individual MoS$_2$ transistors based on 2D nanosheets exfoliated from large-piece single crystal. Inset is the statistical distribution of the mobility values for these transistors. $V_{ds}$ = 1 V. **i** Transmission electron microscopy (TEM) image of an individual MoS$_2$ nanosheet exfoliated from crystal powders. Inset is the corresponding selected-area electron diffraction pattern of this MoS$_2$ nanosheet. Scale bar, 1 μm. **j** X-ray photoelectron spectra (XPS) analysis of the MoS$_2$ nanosheets exfoliated from crystal powders. **k** The thickness distribution of powder-exfoliated MoS$_2$ monolayer nanosheets measured by AFM. Inset is the schematic illustration of an inorganic monolayer capped by organic molecules with a thickness of ~2.8 nm. **l** Raman spectra of 10 individual MoS$_2$ nanosheets. The red dotted lines indicate the position of the $A_{1g}$ and $E_{2g}$ peaks of the Raman spectra.

powders are similar to those from large single crystals (Supplementary Fig. 9). As a signature of high-quality monolayers, the liquid-state photoluminescence from the MoS$_2$ monolayer dispersion in solvent was also observed (Fig. 2c, g). The red-light emission at $\lambda = 661$ nm ($E_g = 1.88$ eV) matches the monolayer bandgap of MoS$_2$ crystal because the bilayer and thicker crystals are expected to exhibit much weaker photoluminescence. By assembling the exfoliated monolayers into solid thin films with thermal annealing at 300 °C (Supplementary Fig. 10), the transistors fabricated from two types of MoS$_2$ nanosheets both deliver carrier mobility of ~7–10 cm$^2$·V$^{-1}$·s$^{-1}$ and current on/off ratio of ~10$^6$ (Fig. 2d, h). This is consistent with the previous report on THAB-exfoliated MoS$_2$ nanosheet thin films[2]. The field-effect mobility values were extracted from the linear region of the $I_d-V_{gs}$ transfer curves. In specific, it is calculated following the equation $\mu = g_m \cdot L / (W \cdot C \cdot V_{ds})$, where $\mu$, $g_m$, $L$, $W$, $C$, and $V_{ds}$ denote field-effect mobility, transconductance, channel length, channel width, gate capacitance, and drain-source voltage, respectively. Also, similar high material and device uniformity was confirmed through the small device-to-device performance variation among 20 individually measured transistors, which is comparable to those previously obtained with large-size single crystals (Fig. 2d, h and Supplementary Fig. 11).

In addition, the intercalation reaction proceeds faster with micron-size crystal powders than with millimeter-size bulk single crystals, which is mostly determined by the surface area and number of crystal boundaries for molecular intercalation. For example, the powder intercalation completes in ~1 h (weight ~100 mg), which is about 2-3 times faster than that using large-size bulk single crystals of similar weight (~3 h). From the transmission electron microscopy (TEM) analysis, the exfoliated MoS$_2$ 2D monolayer is free of structural damage and liquid metal residues (Fig. 2i). The hexagonal electron diffraction pattern matches the (001) crystal plane of MoS$_2$, confirming the intact crystal structure. X-ray photoelectron spectra (XPS) analysis suggests the preservation of the pristine 2H crystal structure and the absence of phase transition to 1T structure throughout the powder-based electrochemical intercalation and exfoliation (Fig. 2j). Also, the Ga scan confirms the absence of gallium residues from the liquid metal. By analyzing 100 individual nanosheets in AFM images, we have obtained a narrow thickness distribution in which >98% are monolayers with a thickness of ~2.8 nm (Fig. 2k). The monolayer structure was also confirmed by X-ray diffraction (XRD) pattern (Supplementary Fig. 12), in which only the organic/inorganic superlattice diffraction peaks ($d$ ~2.8 nm) were observed while the intrinsic MoS$_2$ crystal peak ($d$ ~0.6 nm) is absent. Raman spectra of the exfoliated nanosheets show a consistent wavenumber separation between $E_{2g}$ and $A_{1g}$ peaks of ~19 cm$^{-1}$, matching the value for monolayer crystals instead of bilayer or thicker counterparts (Fig. 2l). All these data suggest that the use of liquid metal in the intercalation and exfoliation reaction, compared with the standard intercalation without liquid metal, does not introduce noticeable changes to the morphology, crystal structure, and surface molecular layer of the exfoliated MoS$_2$ monolayers. Together, using readily available and low-cost micron-size crystal powders as the source material, we have realized high-purity 2D semiconductor monolayers with morphology, electrical performance, and material uniformity that resemble those obtained from standard large-size bulk single crystals.

## A library of 2D electronic material inks

Switching from large-piece bulk single crystal to micron-size powders can greatly expand the scope and applicability of the electrochemical molecular intercalation and exfoliation approach. The high-temperature and time-consuming growth (>100 h) of large single crystals is expensive and requires sophisticated synthetic control. Also these large single crystals may not be available for many 2D materials. By contrast, micron-size crystal powders are much cheaper and readily available for most common 2D layered crystals. As a proof of concept,

we have shown that >50 types of 2D nanosheets can be intercalated and exfoliated from corresponding powders, ranging from TMDs (e.g., ZrS$_2$, NbSe$_2$, and MoTe$_2$), main group metal chalcogenides (e.g., InSe, SnSe$_2$, and Bi$_2$Se$_3$), ternary layered crystals (e.g., MnPS$_3$ and ZnIn$_2$S$_4$), layered oxides (e.g., MoO$_3$ and V$_2$O$_5$), to elemental crystals (e.g., graphene and phosphorene) (Fig. 3 and Supplementary Fig. 13). For most TMDs, the exfoliated nanosheets are high-purity monolayers (monolayer purity >90%), including 2D metals/semi-metals such as TiS$_2$, TiSe$_2$, NbS$_2$, NbSe$_2$, TaS$_2$, and TaSe$_2$, and 2D semiconductors such as MoS$_2$, WS$_2$, ZrS$_2$, ZrSe$_2$, HfS$_2$, and HfSe$_2$. By contrast, many bipolar semiconductors such as MoSe$_2$, MoTe$_2$, WSe$_2$, and VSe$_2$ tend to produce multilayer nanosheets, and the exfoliation yield is also lower. Besides, when intercalating 2D powders without significant crystal volume expansion to induce the automatic material detachment from liquid metal (e.g., MoO$_3$, SnS$_2$, and ZnIn$_2$S$_4$), NaHCO$_3$ was added into liquid metal to create abundant internal micropores and thus higher surface area for accessing electrolyte and alkylammonium intercalants. With the addition of NaHCO$_3$, the intercalation and exfoliation yield for these crystals has been greatly improved by about 10 times. Another interesting point is the surface property of these nanosheets and the related dispersion stability in solution. Most 2D nanosheets, such as MoS$_2$ and MoSe$_2$, require PVP polymer surfactant as a capping agent to minimize restacking in organic solvents after exfoliation. However, PVP is not mandatory for some other crystals. For example, we noticed that InSe, In$_2$Se$_3$, MoO$_3$, SnS$_2$, SnSe$_2$, and ZnIn$_2$S$_4$ monolayers can be well exfoliated and stabilized in pure DMF solvent without the addition of polymer surfactant. The exact origin of this discrepancy is not yet clear. In principle, the alkylammonium functionalized nanosheets are expected to disperse in common organic solvents such as DMF, DMSO, and NMP, considering the high solubility of organic alkylammonium molecules in these polar solvents. However, most exfoliated 2D nanosheets do not form a stable dispersion in these solvents without additional surfactants such as PVP. Therefore, it is likely that the molecular configuration of the alkylammonium molecules and their interaction with the inorganic crystalline lattice may differ among diverse 2D layered crystals.

The electrochemical intercalation of 2D semiconductors with wide bandgap ($E_g > 2.5$ eV) and low electrical conductivity was previously challenging when using large-size bulk single crystals as the source materials due to the difficulty in injecting electrons into the conduction band. Because a respectable electrical conductivity of the bulk crystal itself is the prerequisite for the electrochemical reaction to occur (e.g., MoS$_2$, NbS$_2$, and In$_2$Se$_3$). However, many 2D wide-bandgap semiconductors such as SnS$_2$, GaS, GaSe, MnPS$_3$, and MoO$_3$ ($E_g = 2$-3 eV) exhibit low electrical conductivity and thus no noticeable electrochemical intercalation reaction. In specific, for crystals with a small bandgap such as MoS$_2$ crystal ($E_g = 1.2$–1.9 eV), significant volume expansion throughout the entire crystal and successful intercalation reaction can be observed (Fig. 4a). For SnS$_2$ crystal with a medium bandgap ($E_g = 2.1$ eV), only the top half of the crystal shows obvious volume expansion and the bottom half remains unchanged, signifying a limited intercalation level of the bulk crystal. Therefore, the subsequent exfoliation yield for nanosheets is also low. For 2D semiconductors with an even larger bandgap, such as MoO$_3$ and GaS ($E_g = 2.7$ eV), a negligible sign of intercalation reaction was observed. Occasionally, only a very small part of the crystal near the metal clip is intercalated and expanded. The distinct intercalation behavior among these crystals can be quantitatively determined by the recorded electrochemical current profiles (Fig. 4b). These wide-bandgap semiconductor typically delivers significantly lower electrochemical current and thus less noticeable intercalation reaction than other crystals. For example, the electrochemical current in bulk MoO$_3$ crystal remains negligible even at an intercalation voltage of $-8$ V. This is probably due to the very low electrical conductivity of MoO$_3$ crystal which is also confirmed by the electrical measurement. The intrinsic

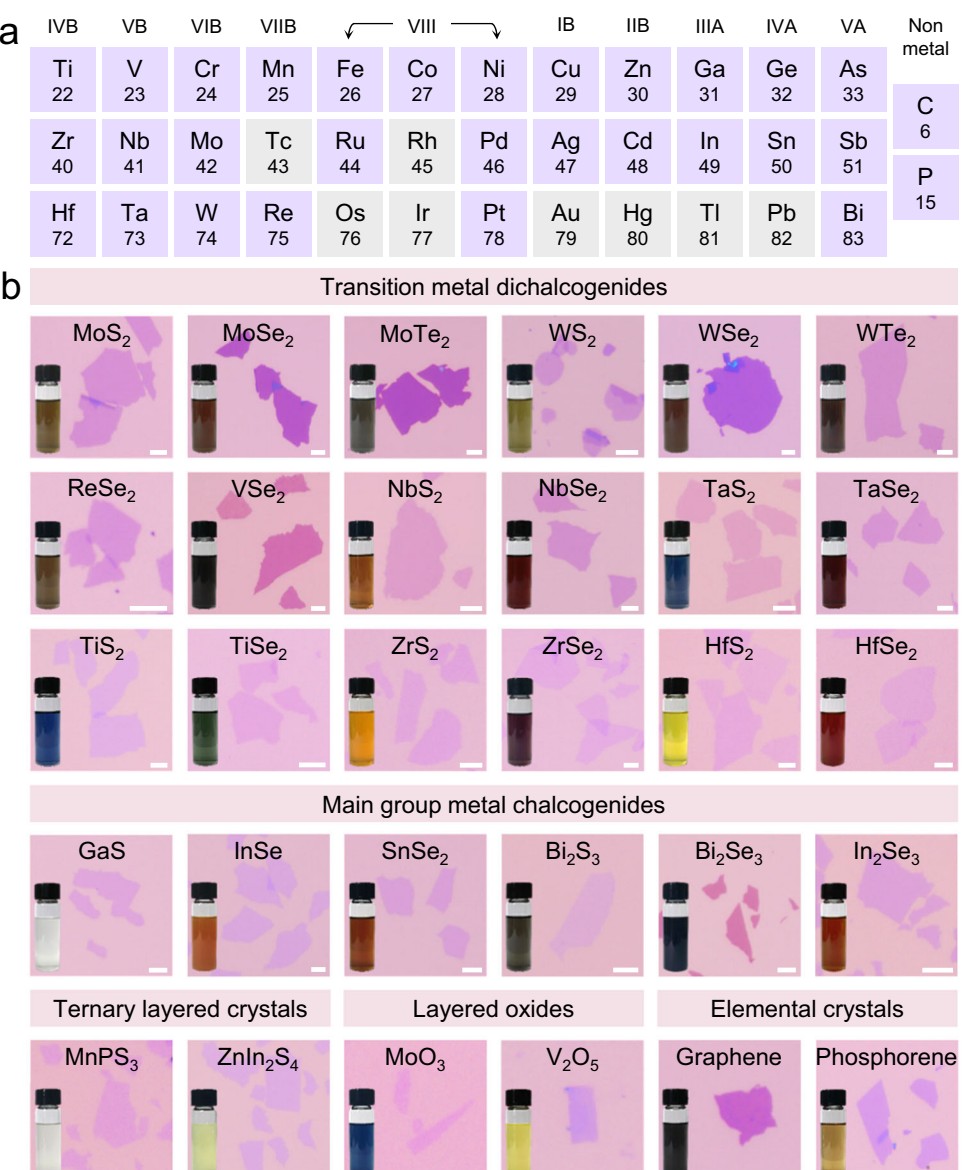

**Fig. 3 | The library of 2D material inks constructed by powder-based electrochemical intercalation and exfoliation. a** Overview of metal and non-metal elements (highlighted in purple) that crystalize into layered crystals with chalcogens, phosphorus, and others (>50 types). These crystals can be used for powder-based electrochemical intercalation and exfoliation into 2D nanosheet inks. **b** Representative optical images of selected 2D nanosheets after exfoliation, ranging from transition metal dichalcogenides (TMDs), main group metal chalcogenides, ternary layered crystals, and layered oxides to elemental crystals. Insets are photographs of each ink solution containing 2D nanosheets dispersed in the solvent. To obtain these large-size nanosheets, the intercalated crystals were exfoliated by manual shaking instead of bath sonication. Scale bars, 5 μm.

$MoO_3$ crystal delivers a negligible current compared with $SnS_2$ and $MoS_2$ at similar thickness, which is smaller by about $10^4$–$10^8$ times (Fig. 4c). The significantly lower electrical conductivity in the wide-bandgap material may result in a larger voltage drop across the crystal and thus smaller effective voltage for intercalation. It prevents the electron injection to the crystal lattice for the intercalation reaction that occurs at the crystal/electrolyte interface.

This challenge can be addressed by the liquid-metal-assisted intercalation approach based on crystal powders. Wide-bandgap semiconductors such as $MoO_3$, GaSe, and GaS can now be electrochemically intercalated and exfoliated into high-quality 2D nanosheet inks. As a proof of concept, we focus on GaS crystal, which has a large bandgap of 2.6–3.0 eV, similar to $MoO_3$. When using a large-piece bulk single crystal, there is no sign of intercalation even at a high intercalation voltage of −20 V (Fig. 4d). By contrast, when mixing the GaS microcrystal powders with liquid metal for electrochemical intercalation, the volume expansion in crystal and color change from light yellow to brown was observed at −7 V, signifying the successful molecular intercalation of alkylammonium cations into the GaS crystal (Fig. 4e). The discrepancy in intercalation behavior mostly comes from the reduced resistance in powder anchored on liquid metal framework comparing with the bulk crystal. In the electrochemical reaction, the electrons need to travel a long distance from the metal clip to the crystal and then the crystal/electrolyte interface for intercalation reaction. For 2D wide-bandgap semiconductors with high resistance, a significant potential drop across the entire bulk crystal (on a millimeter scale) can be observed, and thus no intercalation reaction would occur. However, such potential drop can be mostly eliminated in the liquid-metal-assisted intercalation reaction. Inside the mixture slurry, the micron-size crystal powders form an intimate electrical contact

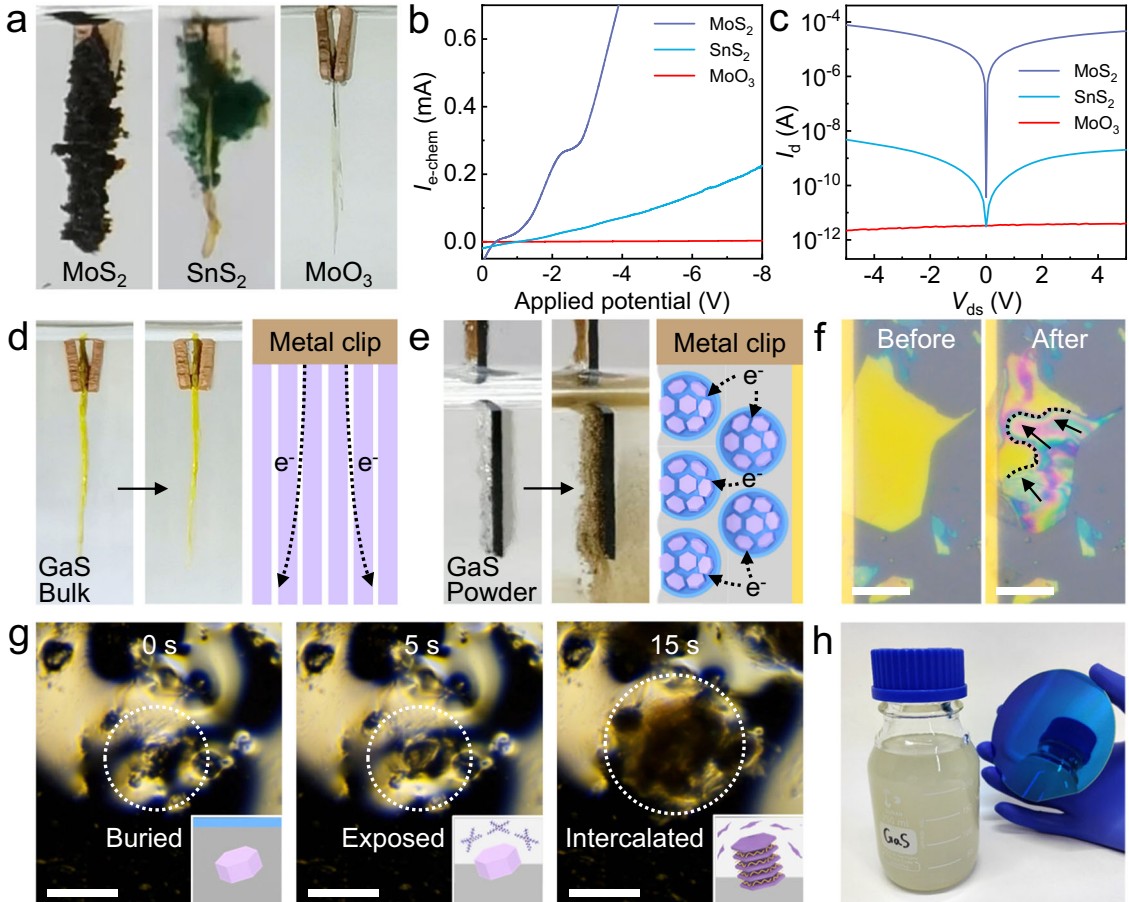

**Fig. 4 | The electrochemical intercalation of 2D wide-bandgap semiconductors enabled by liquid metal. a** Photographs of $MoS_2$, $SnS_2$, and $MoO_3$ single crystals after the electrochemical intercalation, showing different intercalation levels. The length of crystals is ~8 mm. **b** Electrochemical current profiles of the intercalation of $MoS_2$, $SnS_2$, and $MoO_3$ single crystals. **c** Electrical characteristics of intrinsic $MoS_2$, $SnS_2$, and $MoO_3$ thin crystals (~10 nm thickness). **d**, **e** Photographs of GaS crystal before and after intercalation in the form of bulk single crystal (**d**) and powders (**e**). Insets are the schematic illustration of the electron pathway in these two structures. The length of GaS crystal is ~10 mm. **f** Optical microscope images of an individual mechanically exfoliated GaS thin flake (~60 nm thickness) before and after

intercalation. The flake was contacted with a metal electrode for electrical conduction. Scale bars, 100 μm. The dashed line indicates the intercalation front during the reaction, and the arrows indicate the direction of intercalation. **g** In situ dark-field optical images of the slurry surface after intercalation for 0 s, 5 s, and 15 s, showing the gradual exposure of buried microcrystal powders to electrolytes for intercalation. Scale bars, 100 μm. The regions labeled by the dashed lines indicate the location of GaS microcrystals. **h** Photograph of the obtained colloidal solution of GaS 2D nanosheets (200 mL) and the 4-inch thin film deposited on a standard $SiO_2/Si$ wafer.

with the conductive liquid metal framework. The crystal powders (crystal length ~μm) are much smaller than the bulk single crystal (crystal length ~mm), which thus results in a greatly reduced voltage drop. As a result, the electrochemical intercalation of these 2D wide-bandgap semiconductors with low electrical conductivity now becomes feasible.

To further understand the mechanism behind this, we have carried out the intercalation of mechanically exfoliated GaS microcrystal (crystal length ~200 μm, thickness ~60 nm). The electrochemical molecular intercalation of these small GaS microcrystals is confirmed and the advance of molecular intercalation front inside the crystal can also be visualized (Fig. 4f). The morphology evolution of GaS crystal powders in liquid metal mixture slurry during the intercalation can be tracked by in situ optical microscopy measurement. With the applied electrochemical potential, the GaS microcrystals buried in liquid crystal are gradually pushed outward and then exposed to the electrolyte for intercalation reaction (Fig. 4g), as discussed in the previous section. This process was also confirmed by the XRD analysis in which the intensity of intrinsic GaS (001) peak and THAB/GaS superlattice peak both grow at this stage (Supplementary Fig. 14), suggesting the emergence of buried GaS microcrystals on the surface and subsequent

molecular intercalation. Ascribing to the greatly reduced crystal resistance, the recorded electrochemical current grows higher by ~$10^4$ times compared with that using bulk crystal (Supplementary Fig. 15). Following the liquid-metal-assisted intercalation approach using crystal powders, large-scale production of GaS 2D monolayer ink solution and the deposition of uniform 4-inch thin film have been readily achieved (Fig. 4h). These results prove that the electrochemical intercalation reaction of 2D wide-bandgap semiconductors (e.g., GaS, GaSe, and $MoO_3$) that does not occur in previous large-size single crystal can now be enabled by the liquid-metal-assisted method. Importantly, these solution-processable 2D wide-bandgap semiconductors can be used as the dielectric material and further integrated with other 2D semiconductors and 2D metal inks for the fabrication of transistors, memristors, and other electronic devices over a large area and at an affordable cost.

## Solution-processable integration of van der Waals thin films

The creation of a rich library of functional 2D electronic material inks that span from semiconductors, metals, to dielectrics allows for the integration of solution-processable van der Waals thin film electronics. Following the powder-based intercalation and exfoliation approach,

both n-type (e.g., MoS$_2$, MoSe$_2$, and WS$_2$) and p-type (e.g., MoSe$_2$, WSe$_2$, and MoTe$_2$) 2D semiconductor inks and thin films have been prepared. For example, the n-type MoS$_2$, MoSe$_2$, and WS$_2$ thin films assembled from the solution-processable 2D inks deliver an electron mobility of ~5–10 cm$^2$·V$^{-1}$·s$^{-1}$ and a current on/off ratio of 10$^6$ (Fig. 5a, b). On the other hand, MoSe$_2$, WSe$_2$, and MoTe$_2$ typically are bipolar and incline to exhibit n-type transport behavior after exfoliation (Supplementary Fig. 16). But the p-type semiconducting characteristics can be awakened by proper chemical treatment such as bromine for WSe$_2$ nanosheets. The fabricated p-type transistor delivers hole mobility up to ~20 cm$^2$·V$^{-1}$·s$^{-1}$ and a current on/off ratio of 10$^6$ (Fig. 5c). The solution-processable fabrication of both n-type and p-type transistors defines the material foundation for more complex and practical electronics such as complementary metal-oxide-semiconductor (CMOS) devices. In addition to transistors, memory devices such as memristors can also be fabricated with 2D semiconductor inks. Using Pt and Ti electrodes, switching behavior between high and low-resistance states was observed in solution-processable MoS$_2$, MoSe$_2$, and WSe$_2$ thin films (Supplementary Fig. 17). The transition from high-resistance state to

low-resistance state is probably due to the atomic migration of sulfur and selenium defects in TMD crystals under the applied electric field that modulates the Schottky barrier at Pt/TMDs interface[5]. However, the long-term operation stability of the memristors remains a challenge, which requires further optimization of the device fabrication process, such as bottom electrodes with smaller surface roughness and reduced edge spikes.

The metal-semiconductor contact is important for delivering optimized current in the transistor. Following the powder-based intercalation and exfoliation approach, 2D metallic/semi-metallic crystals, including NbS$_2$, NbSe$_2$, TaS$_2$, and others, can be prepared as ink materials. The assembled thin films exhibit high electrical conductivity of ~3000 S/cm at room temperature and can thus be used as the electrode material in transistors (Fig. 5d, e). In particular, stable current can be delivered with NbS$_2$, NbSe$_2$, and TaS$_2$ films of thickness down to ~5 nm (Supplementary Fig. 18). At such a small thickness, the conventional evaporated gold and aluminum metal electrodes might experience high risk of failing due to thickness fluctuation and potential film breakage. However, the 2D crystals can maintain high

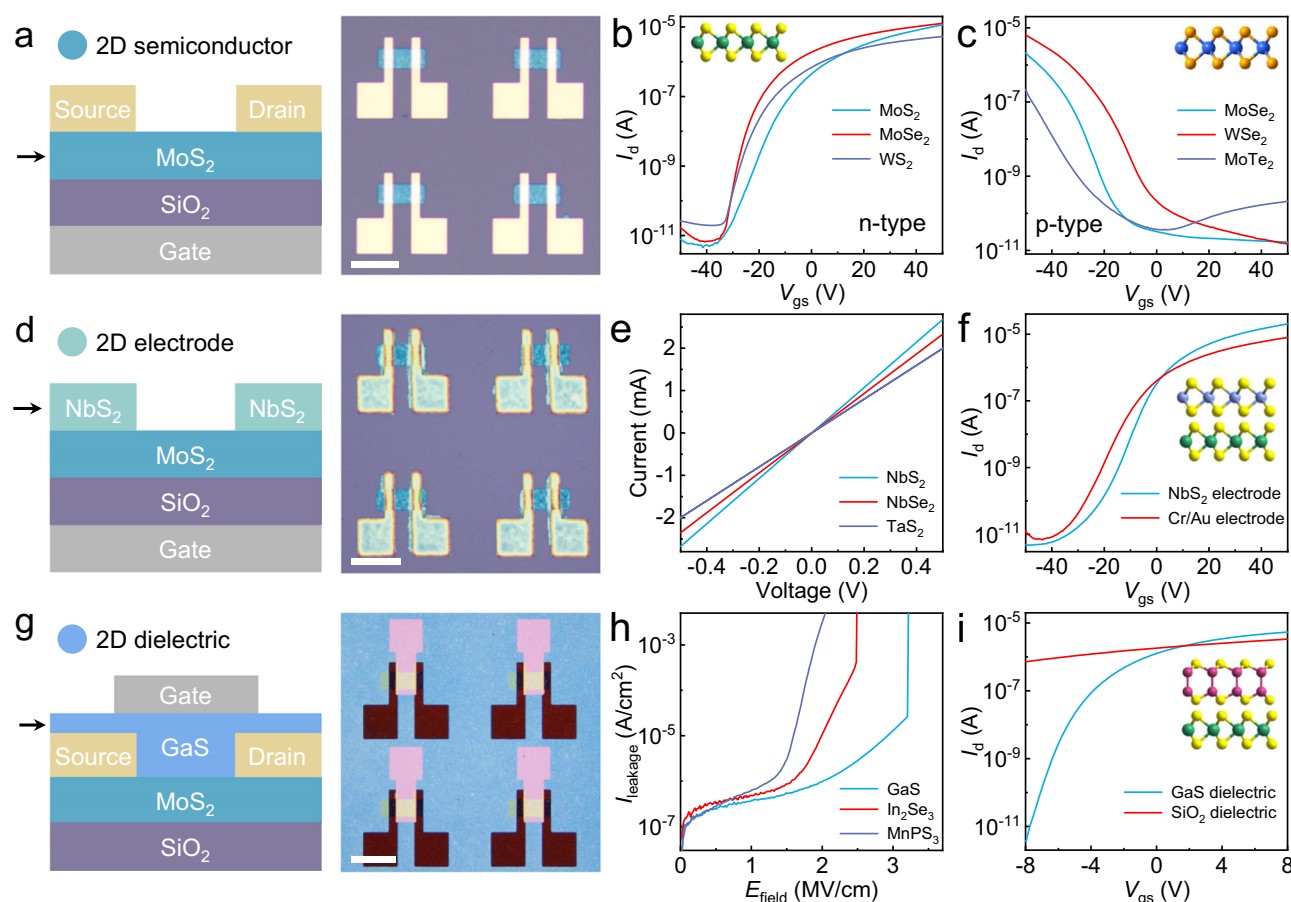

**Fig. 5 | All-solution-processable integration of 2D semiconductors, metals, and dielectrics for van der Waals thin-film transistors. a** Schematic illustration and optical image of solution-processable MoS$_2$ transistors using standard microfabrication techniques. Scale bar, 100 μm. **b, c** Representative $I_d$-$V_{gs}$ transfer characteristics of transistors based on n-type semiconducting MoS$_2$, MoSe$_2$, and WS$_2$ thin films (**b**) and p-type semiconducting MoSe$_2$, WSe$_2$, and MoTe$_2$ thin films (**c**). The substrate is 100 nm SiO$_2$/Si substrate, and the electrode is 20/50 nm Cr/Au for (**b**) and 70 nm Au for (**c**). $V_{ds}$ = 1 V. Insets are the schematic illustration of the crystal structure of MoS$_2$ (**b**) and WSe$_2$ (**c**). **d** Schematic illustration and optical image of transistors by integrating solution-processable 2D metallic NbS$_2$ as contact electrode and MoS$_2$ thin film as a semiconducting channel. Scale bar, 100 μm. **e** Electrical characteristics of NbS$_2$, NbSe$_2$, and TaS$_2$ thin films with thickness of ~15 nm, showing high electrical conductivity. **f** Comparison of $I_d$-$V_{gs}$ transfer characteristics of MoS$_2$ transistors with solution-processable NbS$_2$ and evaporated Cr/Au electrodes. $V_{ds}$ = 1 V. Inset is the schematic illustration of the crystal structure of stacked MoS$_2$ and NbS$_2$. **g** Schematic illustration and optical image of transistors using solution-processable 2D GaS thin film as dielectric and MoS$_2$ thin film as a semiconducting channel. Scale bar, 100 μm. **h** Electrical characterizations of leakage current of GaS, In$_2$Se$_3$, and MnPS$_3$ films. **i** Comparison of $I_d$-$V_{gs}$ transfer characteristics of MoS$_2$ transistors with 40-nm-thick solution-processable 2D GaS and conventional 100-nm-thick thermal SiO$_2$ gate dielectric. $V_{ds}$ = 1 V. Inset is the schematic illustration of the crystal structure of stacked MoS$_2$ and GaS.

material integrity and associated electrical conductivity even within this thickness range. In a typical case study, we have fabricated a thin-film transistor with solution-processable $MoS_2$ semiconductor channel and $NbS_2$ electrode based on successive ink spin coating process (Supplementary Fig. 19). An electron mobility of 10–15 cm²·V⁻¹·s⁻¹ and current on/off ratio of $10^6$ have been extracted from the $I_d$-$V_{gs}$ transfer curves (Fig. 5f). These device metrics are comparable to the transistor based on same solution-processable $MoS_2$ channel but with thermally evaporated Cr/Au metal electrodes (μ ~10 cm²·V⁻¹·s⁻¹ and on/off ratio of $10^6$). It demonstrates the potential of using solution-processable 2D metal inks for the fabrication of high-performance and large-area electronic devices.

The solution-processable 2D dielectrics remain a challenging topic compared with 2D semiconductors and metals[12,45]. Our method offers an alternative pathway to the preparation of high-quality 2D dielectric ink materials for further material integration and device fabrication. Here we use the wide-bandgap GaS crystal as a typical example for device fabrication (Fig. 5g). Following the powder-based intercalation with liquid metal and exfoliation in PVP/DMF, the 2D GaS monolayers capped with PVP polymer molecules were obtained. After film deposition, an organic/inorganic hybrid PVP/GaS superlattice structure can be assembled and used as a dielectric layer without further high-temperature thermal annealing process (Supplementary Fig. 20a). The breakdown field strength is about 2-3 MV/cm with the leakage current falling below $10^{-6}$ A/cm² (Fig. 5h). The measured dielectric constant is ~6.6–7.6 for GaS which is higher than the thermal silicon oxide ($k$ ~3.9) (Supplementary Fig. 20b). Similarly, many other PVP-capped 2D nanosheets such as $In_2Se_3$ and $MnPS_3$ exhibit decent capacitance and may also serve as the dielectric layer. To investigate the potential of these 2D dielectric materials, we have fabricated a $MoS_2$ transistor with a spin-coated 40-nm-thick GaS layer as the top gate dielectric (Supplementary Fig. 21). Here, the GaS dielectric layer was directly used after spin coating without further thermal annealing or chemical treatment. In contrast to the dielectric layer that requires high-temperature annealing or oxidative UV irradiation[12,45,46], this material shows excellent compatibility with most device structures and the current fabrication process in industrial fabrication. Within the applied gate voltage range of − 8 V to 8 V, a current on/off ratio of ~$10^6$ was obtained in the $MoS_2$ transistor, which outperforms that with 100-nm-thick thermal $SiO_2$ in the same voltage range (on/off ratio ~5) (Fig. 5i). This is mostly ascribed to the larger dielectric constant and smaller thickness which together result in the higher capacitance. Also, the gate leakage current remains at ~pA level within this voltage range, which is desired as a good dielectric material (Supplementary Fig. 22). Together, we have proposed solution-processable 2D dielectrics with low processing temperature (<100 °C) while simultaneously delivering high gate performance in 2D transistors. It may offer an alternative pathway to the low-cost fabrication of next-generation large-area flexible and wearable 2D electronics in replacement of traditional oxide dielectrics that are processed by costly sputtering or atomic layer deposition.

## Discussion

In summary, we have reported a general approach for exfoliating a wide range of layered crystals into solution-processable 2D monolayers using low-cost and readily available crystal powders as source material. This is enabled by dispersing these powders in liquid metal, which serves as both the flexible anchoring framework and intimate local electrical contact. After intercalation, the exfoliated $MoS_2$ monolayers obtained from powders exhibit comparable material morphology, crystallinity, and electrical characteristics to those from previous large-piece single crystals. Then, the solution-processable production of >50 types of 2D nanosheets has been demonstrated for the creation of a library of 2D electronic material inks, ranging from transition metal dichalcogenides, main group metal chalcogenides,

ternary layered crystals, layered oxides, to elemental crystals. Furthermore, the liquid-metal-assisted intercalation of crystal powders can significantly lower the intercalation barriers, enabling the production of 2D wide-bandgap semiconductor monolayers (e.g., GaS, GaSe, and $MoO_3$) that are previously challenging with large-size bulk single crystals. With the construction of the functional 2D inks library, the exfoliated metal/semi-metal, semiconductor, and wide-bandgap semiconductor monolayers can be used as solution-processable building blocks of electrodes, semiconductors, and dielectrics for the construction of electronic devices such as transistors and memristors. Together, our study offers an alternative pathway to the production of high-quality 2D semiconductors, metals, and dielectrics from low-cost and readily available crystal powders beyond the pricy bulk single crystals. It may thus push forward the development of cost-effective and large-area fabrication of 2D-material-based electronic devices for diverse technological areas.

## Methods

### Preparation of mixture of liquid metal and 2D powders

The liquid metal of Galinstan was prepared by mixing gallium (68.5%), indium (21.5%), and tin (10.0%). The mixture was prepared in a glove-box and then mechanically stirred for 30 min to make a homogeneous liquid following the established method. Then the selected 2D powders (e.g., $MoS_2$, $Bi_2Se_3$, $In_2Se_3$, $HfS_2$, $SnS_2$, $MoO_3$, and GaS crystal powders) were added to the liquid metal with a powder weight of 17% w/w (the typical range is 5% to 50%, or powder/liquid metal weight ratio of 1:20 to 1:1). The mixture slurry was continuously stirred for several minutes until the silver and shiny appearance was restored. The mixing time and powder/liquid metal ratio vary among different crystal powders. The silver slurry was then spread onto a solid substrate such as gold-plated silicon with a coating thickness of ~2 mm over an area of ~1.5 cm². For some crystals such as $MoO_3$, $SnS_2$, and $ZnIn_2S_4$, 20% w/w $NaHCO_3$ powders can be mixed with the liquid metal to improve the reaction yield. After baking at 150 °C for 10 min in air, $NaHCO_3$ decomposes into a gaseous product and thus results in a porous framework with an enlarged surface area. The slurry-coated substrate was connected to a copper clip and served as the working electrode for the following electrochemical molecular intercalation reaction.

### Electrochemical molecular intercalation

For the electrochemical intercalation, a two-electrode cell was used with the slurry-coated silicon substrate, and a graphite rod was placed as the cathode and anode, respectively. The other equipment setup and intercalation parameters are similar to the standard protocol of using bulk single crystals. The electrolyte contains quaternary ammonium bromide such as tetraheptylammonium bromide (THAB, 98% from TCI) (5 mg/mL or higher) and PVP in 40 mL dimethylfor-mamide. The applied intercalation voltage was typically 5–10 V and the intercalation process was allowed to proceed for 30–60 min. The weight percentage of crystal powders, electrolyte concentration, and intercalation voltage could be tuned to adjust the reaction rate. During the electrochemical intercalation, the colorless solution surrounding the graphite electrode slowly turned yellow due to the formation of $Br_2$. In the meantime, the intercalated crystals in the slurry detached from the liquid metal surface and settled to the bottom of the vial.

### Material exfoliation and 2D inks formulation

The intercalated powders were collected and washed with ethanol before sonication in 30 mL 0.2 M PVP/DMF (PVP: MW ≈ 40,000, Sigma-Aldrich) or pure DMF solution for 5 min. To obtain nanosheets with larger lateral dimensions, 5–10 min of manual shaking can be used for exfoliation instead of bath sonication. For the nanosheets exfoliated in DMF (e.g., $In_2Se_3$, $SnS_2$, and $MoO_3$), the materials were centrifuged and washed with DMF two more times. For nanosheets functionalized with

PVP (e.g., $MoS_2$, $Bi_2Se_3$, and GaS), the obtained materials were washed with IPA to remove the excessive PVP. To get rid of large chunks or other impurities, the final dispersion in DMF or IPA was centrifuged at 2000 rpm for 5 min and the precipitates were discarded. In addition to DMF, many other organic solvents, such as NMP and DMSO, can also be used. For liquid-state photoluminescence measurement, the $MoS_2$ monolayer ink was treated with bis(trifluoromethane)sulfonimide (TFSI) and dispersed in dichloroethane.

### Thin film assembly and post-treatment

The films on the $SiO_2$/Si, glass slide, and plastic substrate were prepared by spin coating the ink solutions. An additional 3000 rpm centrifuge for 5 min was performed to discard any precipitates before adjusting the concentration of the final ink solution. The optical absorbance of the colloidal solution was employed to determine the concentration of the nanosheet dispersion. In the case of $MoS_2$, the peak absorbance at around 436 nm was tuned to be 0.45 (cuvette length = 10 mm) for the solution that was diluted 200 times. The ink solution was spin-coated four times on the 100 nm $SiO_2$/Si substrate at a speed of 2000 rpm for 20 s. The $SiO_2$/Si substrate was pre-cleaned with IPA and treated with oxygen plasma (-10 min) before spin coating. The film can be immersed in a 10 mg/ml TFSI/dichloroethane solution at 70 °C for 1 h to optimize the on/off ratio when using $MoS_2$ nanosheets with PVP. Then the prepared thin film on the substrate was thermally annealed at 200 °C (e.g., for $MoSe_2$, $MoTe_2$, $WS_2$, and $WSe_2$) or 300 °C (e.g., for $MoS_2$, $NbS_2$, $NbSe_2$, and $TaS_2$) for 1 h in a tube furnace with argon (or argon/hydrogen) protection (ramp rate was 12 °C/min). The p-type behavior of $MoSe_2$, $WSe_2$, and $MoTe_2$ can be awakened by immersing the annealed films in $Br_2$/o-dichlorobenzene solution (20–200 μM) for 30 min at 150 °C. The ink concentration and spin coating procedure may be specifically optimized for different ink materials to produce continuous thin films with the desired thickness.

### Device fabrication

The thin-film transistors were fabricated on various substrates following the standard photolithography patterning, dry etching, and thermal evaporation of Cr/Au (20 nm/50 nm) or Au (70 nm) source/drain electrodes. For the memristor device, e-beam evaporation was used for the deposition of Ti/Pt (10 nm/30 nm) and Ti/Pd (10 nm/30 nm) electrodes. For the $MoS_2$ back-gate transistor with $NbS_2$ electrode, e-beam lithography was used to define the source and drain electrodes on the patterned $MoS_2$ film, followed by spin coating and lift-off. After fabrication, the whole device was thermally annealed at 300 °C for 1 h in a tube furnace with an argon gas atmosphere. Then additional TFSI/dichloroethane solution treatment and annealing at 300 °C for 1 h is applied to recover the $MoS_2$ performance. For the $MoS_2$ transistor with GaS top-gate dielectric, Cr/Au (5 nm/20 nm) source/drain electrodes were defined on the patterned $MoS_2$ channels followed by spin coating 40 nm GaS dielectric layer. On top of the GaS film, Au (30–50 nm) gate electrodes were prepared with photolithography patterning and thermal evaporation. Finally, the whole device was soaked in 50 °C chloroform for 30 min and then blown dry.

### Characterizations

Characterizations were carried out using scanning electron microscopy (SEM, Hitachi SU-8010), transmission electron microscopy (TEM, Hitachi H-7700; acceleration voltage, 100 KV. JEOL JEM 2100 F; acceleration voltage, 200 KV), X-ray diffraction (XRD, Bruker D8 ADVANCE X-ray powder diffractometer), atomic force microscopy (AFM, Bruker Dimension Icon Scanning Probe Microscope), Ultraviolet-visible spectrophotometer (UV-visible, Hitachi U-3900), Raman and Photoluminescence spectroscopy (Horiba, 532 nm laser wavelength), and X-ray photoelectron spectroscopy (XPS, PHI Quantro SXM). The electrical measurements of transport characteristics were conducted at room temperature under ambient conditions (in vacuum and dark) with a probe station and Keysight B1500A and 2912B measurement unit.

### Data availability

Relevant data supporting the key findings of this study are available within the article and the Supplementary Information file. All raw data generated during the current study are available from the corresponding authors upon request.

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

## Acknowledgements

Z.L. acknowledges the financial support from the National Natural Science Foundation of China (NSFC, grant No. 22275113), Beijing Natural Science Foundation (grant No. Z240025), and Tsinghua University Dushi Program. Z.S. was supported by ERC-CZ program (project LL2101) from Ministry of Education Youth and Sports (MEYS) and by the project Advanced Functional Nanorobots (reg. No. CZ.02.1.01/0.0/0.0/15_003/0000444 financed by the EFRR). We acknowledge the Cell Biology Facility affiliated with the Center of Biomedical Analysis, Tsinghua University, for technical assistance and equipment support on Hitachi H-7650. We also acknowledge the Center of Nanofabrication, Tsinghua University, for support on AFM, photolithography, and material etching.

## Author contributions

Z.L. designed and supervised the research. S.W. and Z.L. developed the intercalation and exfoliation approach, fabricated the electronic devices, and analyzed the results. S.W. and J.X. collected the Raman and photoluminescence spectra. Y.X. and Y.L. assisted with the synthesis of liquid metal. W.L., J.X., J.G., J.H., J.Hou, H.Z., and Z.S. assisted with the material preparation and device fabrication. S.W. and W.L. performed the electron microscopy analysis on the exfoliated materials. S.W. and Z.L. co-wrote the paper. All authors discussed the results and commented on the manuscript.

## Competing interests

The authors declare no competing interests.
