## [Peer Review File · Nature Communications]

A library of 2D electronic material inks synthesized by liquid-metal-assisted intercalation of crystal powdersReviewer #1 (Remarks to the Author):

This study presents a novel liquid-metal-assisted approach for producing 2D semiconductor inks, enabling large-area and low-cost production of flexible electronic devices. The methodology appears sound, with detailed methods provided for reproducibility. However, further analysis could be beneficial to fully support the conclusions, and clarity on the specific advantages over existing methods in the literature.

1. The authors suggested that the use of liquid metal facilitates molecular intercalation and exfoliation, but liquid metal does not wet the precursor bulk materials. Once the oxide is removed at a reducing potential, the precursor particles would detach locally from the surface and possibly lose electric contact. How would authors justify that those particles would remain electrically connected at the interface.

2. Would the intercalation occur without applying voltage?

3. Control experiments without electric field (potential) are needed. Any exfoliated sheets can be produced without voltage?

Reviewer #2 (Remarks to the Author):

Wang et al report the use of liquid metal to produce a slurry of bulk 2D material crystals that can be easily exfoliated by electro-chemical exfoliation. This enables to avoid use of expensive large bulk crystals. The inks contain large nanosheets with thickness below 3 layers and they are free from metal residuals. The quality of the nanosheets is comparable to those obtained using large crystals, as shown by the mobility and on/off ratio of transistors made of spin coated films of 2D semiconducting nanosheets. The method is suitable for exfoliation of a wide range of 2D materials, including oxides that can be used as dielectrics in transistors.

The authors tackle an important problem in electro-chemical exfoliation of 2D materials beyond graphene, which is related to the electrode preparation when using non-conductive materials. Although liquid metal-assisted production of 2D material has been reported already (see works from one of the co-authors), to the best of the referee knowledge, this method has never been combined with electro-chemical exfoliation. Therefore, there is some degree of novelty in this work, however there is no further advantage as compared to standard electro-chemical exfoliation. The fabrication of the transistor still requires treatment in TFSI followed by annealing at 200 degrees (PL signal is also collected after treatment in acid), so there is no improvement in the quality of the material, which is the real bottleneck for applications.

There is also something unclear on the chemical functionalization of the material, as also stated by the authors. Some nanosheets requires use of surfactants to achieve stable dispersion - this is signature of chemical functionalization of the material, even if not resolved by XPS. This problem seems also to be suggested in previous literature reporting liquid-metal assisted production of 2D materials.

The authors should also show statistical AFM analysis of size distributions obtained of powder vs large crystals. Finally, for this method to be really scalable, the authors should show batch to batch reproducibility.

In conclusion, this work reports an original method to avoid use of large bulk crystals for electro-chemical exfoliation, however it seems more indicated for a specialised journal dedicated to the 2D community, such as npj 2D Materials.

Please note: I co-reviewed this manuscript with one of the reviewers who provided the listed reports. This is part of the Nature Communications initiative to facilitate training in peer review and to provide appropriate recognition for Early Career Researchers who co-review manuscripts."

Reviewer #3 (Remarks to the Author):

I co-reviewed this manuscript with one of the reviewers who provided the listed reports. This is part of the Nature Communications initiative to facilitate training in peer review and to provide appropriate recognition for Early Career Researchers who co-review manuscripts

Reviewer #4 (Remarks to the Author):

The manuscript describes the use of liquid metal Gallinstan to enable the electrochemical intercalation and exfoliation of a broad range of 2D layered materials starting from powders. Using this approach the authors report the liquid exfoliation of 2D metal-like, semiconducting and insulating flakes with promising electrical properties, comparable to those of flakes electrochemically exfoliated from bulk 2D materials. In addition, electrochemical exfoliation of 2D insulating materials becomes possible using a liquid metal/2D material slurry. By tuning the size of the flakes in the powder, the authors obtained 2D flakes with different lateral sizes in the μm range. The obtained dispersions of flakes are combined to fabricate fully solution-processed transistors with mobility up to $10 \text{ cm}^2 \text{ V}^{-1} \text{ s}^{-1}$.

The approach used by the authors is original and interesting, but several issues need to be clarified. I recommend major revisions before considering the manuscript for publication in Nature Communication.

I have the following recommendations:

1) The authors mention that at the end of the intercalation process liquid metal can be used for the next batch of intercalation reaction. An experiment should support this statement. Comparison of the electrical and optical properties of flakes exfoliated using freshly prepared and recycled Gallinstan would be very informative.

2) Which is the V_{sd} for the transfer characteristics shown in Figures 2d and 2h? Is it 1V?

3) Some of the reported mobility values seems overestimated. For completeness the authors should report the used formula to calculate the field effect mobility.

4) In a previous work of some of the present authors (Nature, 562(7726), 254-258, 2018) the average field effect mobility for MoS₂ transistors prepared from flakes electrochemically exfoliated from bulk material was higher than reported in the present work ($10 \text{ cm}^2 \text{ V}^{-1} \text{ s}^{-1}$ on average against $7 \text{ cm}^2 \text{ V}^{-1} \text{ s}^{-1}$ in the present work). Which is the origin of this discrepancy? Are intercalation/exfoliation/flake deposition/flake post-treatments equal in the two works? For instance in the previous work intercalation of bulk MoS₂ was carried out for 1 hour while in the present work (lines 268-269) the authors mention that "the powder intercalation completes in ~ 1 hour (weight $\sim 100 \text{ mg}$) which is about 3 times faster than that of large-size bulk single crystal". It seems that the different intercalation time has an effect on the final device performance. Is it the case? Please comment on these points.

5) How are the performance of FETs prepared with $20 \mu\text{m}$ lateral size MoS₂ flakes as compared to $0.5\text{-}2 \mu\text{m}$ lateral size flakes? When the channel length approaches the average flake lateral size a smaller channel resistance should be expected for the same film thickness.

6) In Figures 2b and f the authors show the AFM analysis of MoS₂ flakes capped with organic ammonium molecules and PVP. However, for efficient flake-to-flake charge transport in the prepared FETs organic contaminants should be removed. Please add the corresponding AFM images after removal of ammonium molecules and PVP. Investigation of the same flake before and after removal of organic contaminants would be very informative.

7) Are Raman/optical reflectivity signals from PVP and ammonium molecules visible in the organic/inorganic superlattices prepared by the authors?

8) The authors mention that a weight ratio of 1:1 to 1:20 for the powder/liquid metal is used. Please specify which weight ratio is used for the MoS₂, NbS₂ and GaS flakes prepared for the fabrication of the transistors reported in Fig. 5.

9) In Figure 5 in the Supplementary Information the authors show a diffractogram for the MoS₂ nanosheets, exhibiting a PVP/THAB superlattice structure. The authors should add the corresponding XRD pattern for the same flakes after removal of PVP/THAB, i.e. for the channel of the prepared FETs.

10) For completeness, in all the transfer characteristics please add the corresponding leakage current.

11) Are the fabricated FETs and memristors stable in air under ambient conditions? Are the memristor curves reproducible after 10, 30 cycles?

12) In Figure 12 of the supporting Information it seems that the 2D semiconductor is deposited on the whole substrate. Isn't it selectively deposited using photolithography?

13) In Figure 13 of the Supporting Information the authors report the XRD pattern of GaS after annealing at 200°C, showing the preserved superlattice structure. However, in a previous work of some of the present authors (Nature, 562(7726), 254-258, 2018) PVP and ammonium molecules were decomposed from a MoS₂ film after annealing at 200 °C. Were the annealing conditions (ramp rate, used gases,..) the same in this previous work and in the present work? Please clarify this point that might be confusing for the readers.

14) In some points, the manuscript would benefit from language editing

Response to Reviewer 1:

General comments. *This study presents a novel liquid-metal-assisted approach for producing 2D semiconductor inks, enabling large-area and low-cost production of flexible electronic devices. The methodology appears sound, with detailed methods provided for reproducibility. However, further analysis could be beneficial to fully support the conclusions, and clarity on the specific advantages over existing methods in the literature.*

Response: We thank the reviewer for carefully reviewing the manuscript and considering our method as “*novel*” and “*sound*”. We especially appreciate the specific questions raised by the reviewer, and welcome the opportunity to address these questions and describe the changes we have made accordingly in the manuscript.

Specific comments.

(1) *The authors suggested that the use of liquid metal facilitates molecular intercalation and exfoliation, but liquid metal does not wet the precursor bulk materials. Once the oxide is removed at a reducing potential, the precursor particles would detach locally from the surface and possibly lose electric contact. How would authors justify that those particles would remain electrically connected at the interface.*

Response: Thanks for the insightful question. In the experiment, we observed the intercalation of crystal powders before their detachment from liquid metal (Fig. 4g). Therefore, it is likely that the electrochemical reduction process of the gallium oxide is slower than the intercalation of 2D microcrystals. In specific, when exposed to the electrolyte, the oxide on top is electrochemically reduced to expose the buried microcrystal for intercalation. In the meantime, the oxide on bottom is not yet reduced and can still connect the microcrystals to the main body of liquid metal. Such an intermediate state when the microcrystal is intercalated but remains connected to the liquid metal has been observed (2nd and 3rd panels in Fig. R1-1a). As a comparison, within the same time (e.g., 15-60 s), the surface oxide of liquid metal is not yet fully reduced which suggests a slower oxide reduction process than the molecular intercalation (Fig. R1-1b). On the other hand, this process can also be verified by the high monolayer purity in the final product, which signifies the full intercalation of these 2D crystal powders before detachment. It suggests that most powders remain the electrical contact with liquid metal during the intercalation process. Otherwise, the non-intercalated crystals are likely to be exfoliated into nonuniform and thick nanosheets instead of high-purity monolayers. Accordingly, we have now added relevant discussion on this point in the revised manuscript.

On page 6 of the revised manuscript: “*When exposed to the electrolyte, the oxide on top is electrochemically reduced to expose the buried microcrystals for intercalation. In the meantime, the oxide on bottom is not yet reduced and can still connect the microcrystal to the main body of liquid metal.*”

Figure R1-1 | The electrochemical intercalation of MoS₂ microcrystals in slurry and the reduction of Galinstan oxide. **a**, *In situ* optical images of the MoS₂ crystals/Galinstan slurry in THA⁺/DMF electrolyte under negative potential for 0 s, 15 s, 60 s, and 180 s. Scale bars, 50 μ m. **b**, *In situ* optical images of electrochemical reduction of surface oxides on Galinstan in THA⁺/DMF electrolyte. Scale bars, 200 μ m.

(2) *Would the intercalation occur without applying voltage?*

Response: Thanks for the question. The intercalation of large-size quaternary alkylammonium molecules such as tetraheptylammonium bromide into MoS₂ crystal is typically non-spontaneous and requires an additional energy input (*Nano Lett.* **2019**, 19, 6819). In our experiment, when exposed to the electrolyte solution without applying potential, both MoS₂ bulk crystal and liquid metal slurry did not show any noticeable sign of intercalation reaction after an hour (Fig. R1-2). By contrast, an intercalation reaction can be observed only when the external potential is applied. This differs from the direct chemical intercalation without external potential in the case of amine and metallic 2D crystals such as aniline into NbS₂ (*Science* **1971**, 174, 493) and methylbenzylamine into TaS₂ (*Nature* **2022**, 606, 902).

Figure R1-2 | The electrochemical intercalation MoS₂ crystal in electrolyte with external voltage. **a,b**, Photograph of a thin piece of MoS₂ single crystal (a) and MoS₂/Galinstan slurry (b) at the original state (left), after immersion in THA⁺/DMF electrolyte for 1 hour (middle), and after immersion in

electrolyte with applied voltage (right).

(3) *Control experiments without electric field (potential) are needed. Any exfoliated sheets can be produced without voltage?*

Response: Thanks for the suggestion. We have now carried out a control experiment without applying external potential. In specific, the MoS₂ powder/Galinstan slurry was soaked in THA⁺/DMF electrolyte for 1 hour in the absence of external potential. No powders were leached into the solution since there was no electrochemical reduction of surface oxide and intercalation reaction (Fig. R1-3a,b). If such mixture slurry was directly sonicated in DMF solvent, the soft liquid metal framework would be broken to release the buried crystal powders into the solvent (Fig. R1-3c). In this case, it will be quite similar to the direct liquid-phase exfoliation (LPE) method without chemical intercalation, which typically yields thick and nonuniform nanosheets (*Science* **2017**, 356, 69; *Science* **2011**, 331, 568). Also, the exfoliation yield is significantly lower than that with ammonium intercalation because most powders remain unexfoliated and precipitate at bottom (Fig. R1-3d). Accordingly, these new data and relevant discussions have also been added to the revised manuscript (Extended Data Fig. 4).

On page 7 of the revised manuscript: “*It is also worth noting that molecular intercalation and Galinstan oxide reduction would not occur in the absence of external potential (Extended Data Fig. 4).*”

Figure R1-3 | Intercalation and exfoliation of MoS₂ crystals with THAB in the absence of external potential. **a,b**, Photographs of MoS₂/Galinstan slurry immersed in THA⁺/DMF electrolyte for 0 hour (a) and 1 hour (b). No sign of intercalation or leaching of powders into solution was observed. **c,d**, Photograph of the obtained colloidal solution (c) and SEM image (d) of the products by directly sonicating the non-intercalated MoS₂/Galinstan slurry in DMF solution. Most of the crystals were not exfoliated and stayed at the bottom of the vial. Scale bar, 1 μm.

Together, we greatly appreciate all the constructive comments from the reviewer, which have motivated us to finely polish the statements and further strengthen our manuscript. According to these insightful suggestions, we have now carefully revised the manuscript, added new data, highlighted all the changes we made, and thus hope to contribute a valuable article to the community.

Response to Reviewer 2 and Reviewer 3 (co-reviewer):

General comments. Wang *et al* report the use of liquid metal to produce a slurry of bulk 2D material crystals that can be easily exfoliated by electro-chemical exfoliation. This enables to avoid use of expensive large bulk crystals. The inks contain large nanosheets with thickness below 3 layers and they are free from metal residuals. The quality of the nanosheets is comparable to those obtained using large crystals, as shown by the mobility and on/off ratio of transistors made of spin coated films of 2D semiconducting nanosheets. The method is suitable for exfoliation of a wide range of 2D materials, including oxides that can be used as dielectrics in transistors.

The authors tackle an important problem in electro-chemical exfoliation of 2D materials beyond graphene, which is related to the electrode preparation when using non-conductive materials.

In conclusion, this work reports an original method to avoid use of large bulk crystals for electro-chemical exfoliation, however it is seems more indicated for a specialised journal dedicated to the 2d community, such as npj 2D Materials.

Please note: I co-reviewed this manuscript with one of the reviewers who provided the listed reports. This is part of the Nature Communications initiative to facilitate training in peer review and to provide appropriate recognition for Early Career Researchers who co-review manuscripts.

(Reviewer 3) I co-reviewed this manuscript with one of the reviewers who provided the listed reports. This is part of the Nature Communications initiative to facilitate training in peer review and to provide appropriate recognition for Early Career Researchers who co-review manuscripts.

Response: We thank both reviewers for carefully reviewing the manuscript and considering our work as “*original method*” and “*tackle an important problem*”. We especially appreciate the specific questions raised by the reviewer, which helped us improve the quality of this work. We welcome the opportunity to address these issues and describe the corresponding changes we have made in the manuscript.

Specific comments.

(1) *Although liquid metal-assisted production of 2D material has been reported already (see works from one of the co-authors), to the best of the referee knowledge, this method has never been combined with electro-chemical exfoliation. Therefore, there is some degree of novelty in this work, however there is no further advantage as compared to standard electro-chemical exfoliation. The fabrication of the transistor still requires treatment in TFSI followed by annealing at 200 degrees (PL signal is also collected after treatment in acid), so there is no improvement in the quality of the material, which is the real bottleneck for applications.*

Response: Thanks for the valuable comment. We are sorry that we have not better elaborated on these key scientific advancements over previous reports and thus have now modified the manuscript accordingly. We agree with the reviewer that the MoS₂ monolayers obtained from the powder-based intercalation approach are similar to those resulting from conventional large-piece single crystals (*Nature* **2018**, 562, 254) in terms of material morphology and electrical

characteristics.

However, this is exactly one of the advantages of our method since 2D semiconductor monolayers resulting from low-cost crystal powders are now of similar quality as those from expensive large single crystals. It **can realize significant cost reduction** of the high-quality solution-processable 2D semiconductor monolayers. Furthermore, compared with the conventional approach using bulk single crystal, this new method features other key scientific advancements that are summarized below.

(1) By avoiding the use of large-piece single crystals, this powder-based method is a universal approach and can now **create a rich library of 2D electronic ink materials (>50 types)**, ranging from transition and main group metal chalcogenides (*e.g.*, MoS₂, ZrS₂, InSe), ternary layered crystals (*e.g.*, ZnIn₂S₄, MnPS₃), layered oxides (*e.g.*, GaS, MoO₃), to elemental crystals (*e.g.*, graphene, black phosphorous). This was **challenging in the previous molecular intercalation approach** due to the difficulty in growing large-piece single crystals of some materials.

(2) The electrochemical molecular intercalation and exfoliation of **high-purity 2D wide-bandgap semiconductor monolayers such as GaS, GaSe, and MoO₃ were previously impossible** due to their low electrical conductivity. The new method using crystal powders and liquid metal can address this challenge and thus greatly expand the scope of electrochemical molecular intercalation/exfoliation chemistry.

(3) Especially, this new method **can now enable the solution-processable 2D monolayer dielectric inks** (*e.g.*, GaS), which deliver more efficient gate control compared with conventional oxide dielectric. More importantly, we have demonstrated the **low-cost solution-processable integration** of 2D semiconductors with 2D conductors and 2D dielectrics, offering exciting potential for the fabrication of large-area all-solution-processable thin-film transistors and memristors.

Together, these new findings and data as well as the material cost reduction represent significant scientific advancements compared with the previous reports and thus will contribute to the research community. And it may further push forward the practical application of solution-processable 2D electronic materials in diverse technological areas at low cost and over large area.

(2) There is also something unclear on the chemical functionalization of the material, as also stated by the authors. Some nanosheets requires use of surfactants to achieve stable dispersion - this is signature of chemical functionalization of the material, even if not resolved by XPS. This problem seems also to be suggested in previous literature reporting liquid-metal assisted production of 2D materials.

Response: Thanks for pointing out this concern, and we are sorry that we had not more clearly elaborated on the surfactants/ligands on the exfoliated nanosheet. For the electrochemical molecular intercalation and exfoliation of 2D bulk single crystals using organic ammonium cations, it has been previously reported that some materials require additional polymer surfactant to achieve stable dispersion while some others do not (*Nat. Protoc.* **2023**, 18, 2814; *Nat. Mater.* **2021**, 20, 181; *Chem* **2021**, 7, 1887). The interaction between organic molecules and exfoliated 2D crystals has been studied in these literature reports. Therefore, the use of

PVP surfactant is common in the electrochemical molecular intercalation/exfoliation and thus not specifically caused by the liquid metal used in this work. For the 2D nanosheets resulting from liquid-metal-assisted intercalation in this work, we would expect similar interaction and mechanism as these reports.

In the previous studies, THAB and PVP were believed to interact with 2D nanosheets via van der Waals force or electrostatic attraction. Because, the dangling-bond-free 2D crystals are not expected to form a large number of strong covalent bonds with either THAB or PVP molecules, which was confirmed by the experimental results and theoretical calculations (*Nature* **2018**, 555, 231). In specific, for the exfoliation without polymer surfactants such as In_2Se_3 and NbSe_2 , the organic THAB ammonium molecular layers are present on the surface of exfoliated nanosheets as the capping ligand (*e.g.*, THAB/ In_2Se_3). For the other materials that require additional PVP as surfactant such as MoS_2 and WSe_2 , the exfoliated nanosheets are capped with an additional polymer layer (*e.g.*, PVP/THAB/ MoS_2 or PVP/ MoS_2). The origin of such discrepancy is probably the weaker interaction force between THAB and some 2D crystals. When these weakly bound THAB molecules detach from the crystal surface, nanosheet aggregation and/or unsuccessful exfoliation are observed. In this case, additional PVP polymer surfactant can bind to the crystal surface to achieve stable dispersion.

By contrast, previous production of non-layered nanocrystals in the 2D form assisted with liquid metal seems to differ from the case reported in our work (*e.g.*, SnO_x in *Nano Lett.* **2020**, 20, 2916; Ga_2O_3 , HfO_2 , and Al_2O_3 in *Science* **2017**, 358, 332; MnO_2 in *Adv. Funct. Mater.* **2019**, 29, 1901649). These nanocrystals, although in the 2D form, are not intrinsic layered compounds or connected by bonding-free van der Waals interaction. Therefore, the crystal surface is full of rich defects and dangling bonds which allow for chemical functionalization through covalent bonds by various polymers, ionic ligands, and hydroxyl groups (*Adv. Funct. Mater.* **2021**, 31, 2010320), which, however, are not available in 2D van der Waals crystals. To avoid potential misunderstanding of this point, we have now added relevant discussion in the revised manuscript.

On page 10 of the revised manuscript: “*All these data suggest that the use of liquid metal in the intercalation and exfoliation reaction, compared with the standard intercalation without liquid metal, does not introduce noticeable changes to the morphology, crystal structure, and surface molecular layer of the exfoliated MoS_2 monolayers.*”

(3) *The authors should also show statistical AFM analysis of size distributions obtained of powder vs large crystals. Finally, for this method to be really scalable, the authors should show batch to batch reproducibility.*

Response: Thanks for the helpful suggestions. Accordingly, we have now added statistical analyses of the lateral size and thickness of MoS_2 nanosheets exfoliated from crystal powders and large single crystals, respectively (Fig. R2-1). Both types of exfoliated MoS_2 nanosheets show an average lateral size of $\sim 0.8\text{-}1.0\ \mu\text{m}$ and a uniform thickness of $\sim 2.8\ \text{nm}$. Accordingly, these new data have now been added to the revised manuscript (Extended Data Fig. 9).

Figure R2-1 | AFM analysis of the exfoliated MoS₂ nanosheets from crystal powders and bulk single crystal. **a-c**, AFM image (a) and statistical analyses of lateral size (b) and thickness (c) of MoS₂ nanosheets exfoliated from crystal powders. Scale bar, 2 μm. **d-f**, AFM image (d) and statistical analyses of lateral size (e) and thickness (f) of MoS₂ nanosheets exfoliated from bulk single crystals. Scale bar, 2 μm.

Figure R2-2 | AFM analyses of the 2D MoS₂ nanosheets obtained from six independent intercalation and exfoliation processes. **a-f**, AFM image (left) and distribution of lateral size and thickness (right) of exfoliated MoS₂ nanosheets prepared in six independent batches. The consistent nanosheet morphology suggests good batch-to-batch reproducibility. Scale bars, 2 μm.

Figure R2-3 | The recyclability of liquid metal for the electrochemical intercalation. **a-d**, Photographs of Galinstan and slurry (a), liquid-state photoluminescence spectrum of the obtained ink solution (b), Raman spectrum of the MoS₂ monolayer (c), and I_d-V_{gs} transfer characteristics of the prepared film (d), using fresh liquid metal for the 1st cycle of intercalation and exfoliation of MoS₂ powders. **e-h**, Photographs of Galinstan and slurry (e), liquid-state photoluminescence spectrum of the obtained ink solution (f), Raman spectrum of the MoS₂ monolayer (g), and I_d-V_{gs} transfer characteristics of the prepared film (h), using recycled liquid metal for the 2nd cycle of intercalation and exfoliation of MoS₂ powders. **i-l**, Photographs of Galinstan and slurry (i), liquid-state photoluminescence spectrum of the obtained ink solution (j), Raman spectrum of the MoS₂ monolayer (k), and I_d-V_{gs} transfer characteristics of the prepared film (l), using re-recycled liquid metal for the 3rd cycle of intercalation and exfoliation of MoS₂ powders. The applied V_{ds} is 1 V for all transfer curves.

Also, we agree with the reviewer about the claim of scalability and thus have tested the batch-to-batch reproducibility. In specific, the intercalation and exfoliation process of MoS₂ nanosheets have been carried out repeatedly by using different crystal powders and liquid metal. According to 6 independent experiments, all the exfoliated MoS₂ nanosheets show a similar distribution of lateral size and thickness (Fig. R2-2), indicating good reproducibility of the powder-based intercalation method among different batches. In addition, the liquid metal can also be recycled for intercalation reaction and the produced MoS₂ monolayers exhibit consistent morphology and quality (Fig. R2-3). Accordingly, we have updated the revised manuscript to include these new data (Extended Data Fig. 5 and 8).

On page 7 of the revised manuscript: “Importantly, the recovered liquid metal can be collected and then mixed with new powders for the next batch of intercalation (Extended Data Fig. 5).”

On page 9 of the revised manuscript: “And the morphology of the monolayer nanosheets is reproducible from batch to batch (Extended Data Fig. 8).”

On page 9 of the revised manuscript: “Therefore, the lateral size and thickness of the MoS₂ monolayer nanosheets exfoliated from crystal powders are similar to these from large single crystals (Extended Data Fig. 9).”

Together, we greatly appreciate these constructive comments from the reviewers, which

have motivated us to finely polish the statements and further strengthen our manuscript. According to those insightful suggestions, we have now carefully revised the manuscript, highlighted all the changes we made, and thus hope to contribute a valuable article to the community.

Response to Reviewer 4:

General comments. *The manuscript describes the use of liquid metal Galinstan to enable the electrochemical intercalation and exfoliation of a broad range of 2D layered materials starting from powders. Using this approach the authors report the liquid exfoliation of 2D metal-like, semiconducting and insulating flakes with promising electrical properties, comparable to those of flakes electrochemically exfoliated from bulk 2D materials. In addition, electrochemical exfoliation of 2D insulating materials becomes possible using a liquid metal/2D material slurry. By tuning the size of the flakes in the powder, the authors obtained 2D flakes with different lateral sizes in the μm range. The obtained dispersions of flakes are combined to fabricate fully solution-processed transistors with mobility up to $10\text{ cm}^2\text{ V}^{-1}\text{ s}^{-2}$.*

The approach used by the authors is original and interesting, but several issues need to be clarified. I recommend major revisions before considering the manuscript for publication in Nature Communication.

I have the following recommendations:

Response: We thank the reviewer for carefully reviewing the manuscript and considering our approach as “*original and interesting*” and for the support of the publication on *Nature Communications*. We especially appreciate the specific questions raised by the reviewer, and welcome the opportunity to address these questions and describe the changes we have made accordingly in the manuscript.

Specific comments.

(1) *The authors mention that at the end of the intercalation process liquid metal can be used for the next batch of intercalation reaction. An experiment should support this statement. Comparison of the electrical and optical properties of flakes exfoliated using freshly prepared and recycled Galinstan would be very informative.*

Response: Thanks for the good suggestion to further strengthen the claim in our manuscript. We agree with the reviewer and have now added the experimental results on recycling the use of liquid metal in the revised manuscript. Specifically, at the end of the intercalation reaction, the remaining liquid metal was collected and then mixed with new MoS_2 powders for the next cycle of reaction. The recycled liquid metal can form a similar slurry mixture with MoS_2 powders (Fig. R4-1). The exfoliated MoS_2 nanosheets using recycled liquid metal exhibit similar monolayer purity (>98%), liquid-state photoluminescence (monolayer emission at $E=1.9\text{ eV}$), Raman characteristics, and electrical performance (*e.g.*, carrier mobility and on/off ratio), resembling the materials obtained from the initial fresh Galinstan. Therefore, these new data can confirm the recyclability of liquid metal in the process.

We greatly appreciate the reviewer’s suggestion to further clarify this point in the manuscript. Accordingly, we have now added these new data and relevant discussions in the revised manuscript (Extended Data Fig. 5).

Figure R4-1 | The recyclability of liquid metal for the electrochemical intercalation. **a-d**, Photographs of Galinstan and slurry (a), liquid-state photoluminescence spectrum of the obtained ink solution (b), Raman spectrum of the MoS₂ monolayer (c), and I_d-V_{gs} transfer characteristics of the prepared film (d), using fresh liquid metal for the 1st cycle of intercalation and exfoliation of MoS₂ powders. **e-h**, Photographs of Galinstan and slurry (e), liquid-state photoluminescence spectrum of the obtained ink solution (f), Raman spectrum of the MoS₂ monolayer (g), and I_d-V_{gs} transfer characteristics of the prepared film (h), using recycled liquid metal for the 2nd cycle of intercalation and exfoliation of MoS₂ powders. **i-l**, Photographs of Galinstan and slurry (i), liquid-state photoluminescence spectrum of the obtained ink solution (j), Raman spectrum of the MoS₂ monolayer (k), and I_d-V_{gs} transfer characteristics of the prepared film (l), using re-recycled liquid metal for the 3rd cycle of intercalation and exfoliation of MoS₂ powders. The applied V_{ds} is 1 V for all transfer curves.

(2) Which is the V_{sd} for the transfer characteristics shown in Figures 2d and 2h? Is it 1V?

Response: Thanks for pointing out the missing information here. The V_{ds} is 1 V for all the transfer curves in Fig. 2d and Fig. 2h. The figure caption has now been updated in the revised manuscript.

(3) Some of the reported mobility values seems overestimated. For completeness the authors should report the used formula to calculate the field effect mobility.

Response: Thanks for the valuable suggestion. We calculated the field-effect mobility values in the linear region of the I_d-V_{gs} transfer curves. In specific, it is calculated following the equation $\mu = g_m \cdot L / (W \cdot C \cdot V_{ds})$, where μ , g_m , L, W, C, and V_{ds} denote field-effect mobility, transconductance, channel length, channel width, gate capacitance, and drain-source voltage, respectively.

Below we will elaborate on the mobility calculation in the manuscript. For example, in the I_d-V_{gs} transfer curves of Fig. 2d,h, the drain-source voltage (V_{ds}) is 1 V and the channel length (L) and width (W) of the transistor are both 40 μ m (Fig. R4-2a,d). The dielectric layer is 100 nm silicon oxide with a gate capacitance (C) of 34.5 nF·cm⁻². Based on the measured transfer

curves and the above parameters, we can calculate the mobility values at various gate voltages (Fig. R4-2). Then the peak mobility of $\sim 9 \text{ cm}^2 \cdot \text{V}^{-1} \cdot \text{s}^{-1}$ and $\sim 7 \text{ cm}^2 \cdot \text{V}^{-1} \cdot \text{s}^{-1}$ were extracted from these graphs. Accordingly, we have now added the equation and relevant discussion in the revised manuscript.

On page 9 of the revised manuscript: “The field-effect mobility values were extracted from the linear region of the I_d - V_{gs} transfer curves. In specific, it is calculated following the equation $\mu = g_m \cdot L / (W \cdot C \cdot V_{ds})$, where μ , g_m , L , W , C , and V_{ds} denote field-effect mobility, transconductance, channel length, channel width, gate capacitance, and drain-source voltage, respectively.”

Figure R4-2 | Calculation of field-effect mobility of the MoS₂ transistor. **a-c**, Optical image (a), I_d - V_{gs} transfer characteristics (b) of the MoS₂ transistor of nanosheets exfoliated from crystal powders, and plot of mobility values at various gate voltages (c). **d-f**, Optical image (d), I_d - V_{gs} transfer characteristics (e) of the MoS₂ transistor of nanosheets exfoliated from bulk single crystals, and plot of mobility values at various gate voltages (f). The applied V_{ds} is 1 V. Scale bars, 50 μm . The channel length and width are both 40 μm . The dielectric layer is 100 nm SiO₂ and the source/drain electrode is 20/50 nm Cr/Au.

(4) In a previous work of some of the present authors (*Nature*, 562(7726), 254-258, 2018) the average field effect mobility for MoS₂ transistors prepared from flakes electrochemically exfoliated from bulk material was higher than reported in the present work ($10 \text{ cm}^2 \text{ V}^{-1} \text{ s}^{-2}$ on average against $7 \text{ cm}^2 \text{ V}^{-1} \text{ s}^{-2}$ in the present work). Which is the origin of this discrepancy? Are intercalation/exfoliation/flake deposition/flake post-treatments equal in the two works? For instance in the previous work intercalation of bulk MoS₂ was carried out for 1 hour while in the present work (lines 268-269) the authors mention that “the powder intercalation completes in ~ 1 hour (weight ~ 100 mg) which is about 3 times faster than that of large-size bulk single crystal”. It seems that the different intercalation time has an effect on the final device performance. Is it the case? Please comment on these points.

Response: Thanks for the good question. Compared with the previous report (*Nature* 2018, 562, 254), the intercalation/exfoliation condition, film deposition, post-treatments, and device fabrication procedure for the bulk MoS₂ crystal in the current work remain nearly identical. In the 2018 paper, the MoS₂ films annealed at 200 $^{\circ}\text{C}$ and 300 $^{\circ}\text{C}$ deliver carrier mobility of $\sim 10 \text{ cm}^2 \cdot \text{V}^{-1} \cdot \text{s}^{-1}$ and $\sim 7 \text{ cm}^2 \cdot \text{V}^{-1} \cdot \text{s}^{-1}$, respectively. The annealing temperature affects the device

characteristics such as carrier mobility and on/off ratio. In the current paper, we annealed the MoS₂ film at 300 °C with a measured mobility of $\sim 7 \text{ cm}^2 \cdot \text{V}^{-1} \cdot \text{s}^{-1}$. So, it is consistent with the value for 300 °C annealing in the literature. If the annealing temperature is decreased to 200 °C, a higher carrier mobility can also be obtained. In addition, the mobility values for the transistors on the same substrate might have a distribution range of $\pm 2 \text{ cm}^2 \cdot \text{V}^{-1} \cdot \text{s}^{-1}$ as seen in both current work (Fig. 2d,h) and previous literature. Therefore, statistical analysis is important for evaluating the device performance between different samples. On the other hand, the duration of the intercalation reaction might affect the yield of the final product. But, as far as we know, the electrical performance of the MoS₂ nanosheets remains quite similar (e.g., intercalated for 2 hours or 5 hours).

To avoid potential misunderstanding of the reported mobility values, we have now added relevant descriptions of annealing temperature and comparison with literature values in the revised manuscript.

On page 9 of the revised manuscript: “By assembling the exfoliated monolayers into solid thin films with thermal annealing at 300 °C (Extended Data Fig. 10), the transistors fabricated from two types of MoS₂ nanosheets both deliver carrier mobility of $\sim 7\text{-}10 \text{ cm}^2 \cdot \text{V}^{-1} \cdot \text{s}^{-1}$ and current on/off ratio of $\sim 10^6$ (Fig. 2d,h). This is consistent with the previous report on THAB-exfoliated MoS₂ nanosheet thin films².”

Figure R4-3 | Mobility values for transistors based on the films constructed by the exfoliated MoS₂ nanosheets. a, The statistical distribution of mobility values for 50 individual transistors annealed at 200 °C (purple bars) and 300 °C (grey bars). The graph is adapted from previous literature (*Nature* **2018**, 562, 254). b, I_d - V_{gs} transfer characteristics of 20 individual MoS₂ transistors annealed at 300 °C. Inset is the statistical distribution of the mobility values for these transistors. $V_{ds}=1 \text{ V}$.

(5) How are the performance of FETs prepared with 20 μm lateral size MoS₂ flakes as compared to 0.5-2 μm lateral size flakes? When the channel length approaches the average flake lateral size a smaller channel resistance should be expected for the same film thickness.

Response: Thanks for the great point. We agree with the reviewer about the influence of nanosheet size on electrical performance. Here we have added the data of transistors using larger MoS₂ nanosheets (lateral size $\sim 10\text{-}20 \mu\text{m}$) as the channel material (Fig. R4-4). The carrier mobility falls in the range of $12\text{-}15 \text{ cm}^2 \cdot \text{V}^{-1} \cdot \text{s}^{-1}$, which is higher than $7\text{-}10 \text{ cm}^2 \cdot \text{V}^{-1} \cdot \text{s}^{-1}$ for the regular MoS₂ nanosheets (lateral size $\sim 0.5\text{-}2 \mu\text{m}$). The increased mobility is probably due to the reduced number of grain boundaries and associated carrier scattering by using large nanosheets. Also, a smaller resistance of the film using large nanosheets is confirmed by the higher current at zero gate voltage, which matches the reviewer’s expectation (Fig. R4-4b,d). However, the large nanosheets tend to be deposited into less uniform films due to the oversized length/width. By contrast, the smaller nanosheets show better material uniformity in the

deposited film using the current recipe. Therefore, the deposition of uniform thin films using large nanosheets might require further optimization of the ink formulation and coating procedures.

Figure R4-4 | Transistors fabricated from MoS₂ nanosheets of different sizes. **a**, Optical image of an individual MoS₂ transistor using large nanosheets with lateral size of 10-20 μm. Scale bar, 10 μm. **b**, I_d - V_{gs} transfer characteristics of the transistor shown in (a). $V_{ds}=1$ V. **c**, Optical image of an individual MoS₂ transistor using small nanosheets with lateral size of 0.5-2 μm. Scale bar, 10 μm. **d**, I_d - V_{gs} transfer characteristics of the transistor shown in (c). $V_{ds}=1$ V.

(6) In Figures 2b and f the authors show the AFM analysis of MoS₂ flakes capped with organic ammonium molecules and PVP. However, for efficient flake-to-flake charge transport in the prepared FETs organic contaminants should be removed. Please add the corresponding AFM images after removal of ammonium molecules and PVP. Investigation of the same flake before and after removal of organic contaminants would be very informative.

Response: Thanks for the helpful suggestion. We have now carried out the AFM study on the same MoS₂ monolayers before and after removing the ammonium and PVP molecules. The monolayer nature of the exfoliated MoS₂ nanosheets was confirmed by the Raman spectra for the consistent wavenumber separation between E_{2g} and A_{1g} peaks of ~19 cm⁻¹ (Fig. 2l). Compared with the pristine nanosheets, the thickness decreased from 2.8 nm to 1.2 nm after annealing (Fig. R4-5a,b), indicating the removal of organic molecules from MoS₂ crystal. It is worth noting that the annealed MoS₂ nanosheets do not show the ideal monolayer thickness of ~0.6 nm. This is probably due to the different tip-sample interactions and potential organic residues that might reside at the interface between nanosheets and substrate. The larger apparent thickness in AFM has been previously observed in MoS₂ and other 2D monolayers (*Nat. Commun.* **2015**, 6, 7817; *ACS Nano* **2013**, 7, 11333; *Nat. Rev. Chem.* **2013**, 5, 263).

More importantly, macroscopic XRD patterns and microscopic cross-sectional TEM analysis can also provide convincing data on the removal of organic molecules. The disappearance of superlattice periodicity (~2.8 nm) and the emergence of pristine MoS₂ lattice (~0.6 nm) indicate the complete removal of the organic molecules between inorganic crystals

(Fig. R4-5c). Since the XRD patterns were collected from a large-area sample (*i.e.*, 2*10 mm), it indicates the macroscopic uniformity of the film after removing organic molecules. On the other hand, we have also carried out a cross-sectional TEM study on the assembled MoS₂ thin film to examine the microscopic structure after the removal of organic molecules (Fig. R4-5d,e). The interlayer distance of ~0.6 nm matches that in the pristine bulk crystal and thus suggests the complete removal of organic molecules between MoS₂ monolayers. Together, the removal of organic molecules from 2D monolayers after thermal annealing can now be confirmed. We appreciate the suggestion from the reviewer and have also added the new data and relevant discussions to the revised manuscript (Extended Data Fig. 7, 10, and 12).

Figure R4-5 | Thickness characterization of MoS₂ monolayers before and after annealing. **a,b**, AFM images of MoS₂ monolayer nanosheets before (a) and after (b) annealing to show the removal of ligands. The apparent thickness in AFM (~1.2 nm) is larger than the monolayer value (~0.6 nm) which may be from the different tip-sample interaction and organic residues at nanosheet/substrate interface. Scale bars, 2 μ m. **c**, XRD patterns of MoS₂ thin film before and after removing organic molecules. **d,e**, Cross-sectional TEM images of the annealed MoS₂ films assembled from exfoliated monolayers. The interlayer distance is ~0.6 nm which matches that in the pristine bulk crystal and thus suggests the complete removal of organic molecules between MoS₂ monolayers. Scale bars, 5 nm (e) and 1 nm (f).

(7) Are Raman/optical reflectivity signals from PVP and ammonium molecules visible in the organic/inorganic superlattices prepared by the authors?

Response: Thanks for the suggestion. Accordingly, we have now collected the FT-IR spectrum of the hybrid organic/inorganic PVP/MoS₂ nanosheet (Fig. R4-6). The vibration peaks at ~2950 cm⁻¹ (C-H stretching) and ~1650 cm⁻¹ (C=O stretching) suggest the existence of THAB and PVP molecules on the exfoliated nanosheets. Similarly, the existence of THAB molecules on exfoliated In₂Se₃ nanosheets was also confirmed by FT-IR in the literature (*Chem* **2021**, *7*, 1887). On the other hand, the Raman signal from these surface molecules appears rather weak and thus was not identified on the nanosheets. More importantly, the signature Raman signal for PVP (~3000 cm⁻¹) overlaps with the PL region of MoS₂ crystal, which may further complicate the identification of organic molecules.

Figure R4-6 | FT-IR spectra collected from pristine MoS₂ powders, THAB, PVP, and organic/inorganic PVP/MoS₂ superlattice.

(8) *The authors mention that a weight ratio of 1:1 to 1:20 for the powder/liquid metal is used. Please specify which weight ratio is used for the MoS₂, NbS₂ and GaS flakes prepared for the fabrication of the transistors reported in Fig. 5.*

Response: Thanks for pointing out the missing information here. We used a powder/Galinstan ratio of 1:5 for MoS₂, NbS₂, and GaS powders which is the typical ratio used for most 2D powders. Increasing the portion of crystal powders (e.g., to 1:2) can result in the production of more exfoliated nanosheets in each batch because more materials are now present in the slurry mixture. Based on our experience, the weight ratio in the range 1:1 and 1:20 can usually work in most scenarios and thus it is suggested in our process. Accordingly, we have now updated the method section in the revised manuscript to clarify this point.

On page 19 of the revised manuscript: “Then the selected 2D powders (e.g., MoS₂, Bi₂Se₃, In₂Se₃, HfS₂, SnS₂, MoO₃, and GaS crystal powders) were added to the liquid metal with a powder/liquid metal weight ratio of 1:5 (the typical range is 1:1 to 1:20).”

(9) *In Figure 5 in the Supplementary Information the authors show a diffractogram for the MoS₂ nanosheets, exhibiting a PVP/THAB superlattice structure. The authors should add the corresponding XRD pattern for the same flakes after removal of PVP/THAB, i.e. for the channel of the prepared FETs.*

Response: Thanks for the suggestion. We have now performed XRD analyses on the same film before and after the removal of organic molecules. The interlayer distance shrinks from 2.8 nm in the hybrid superlattice to 0.6 nm in the recovered pure MoS₂ thin film (Fig. R4-7a). In the meantime, the color of the film at the same location becomes lighter which also confirms the reduced thickness (Fig. R4-7b,c). Accordingly, we have now added these new data in the revised manuscript (Extended Data Fig. 12).

Figure R4-7 | Characterization of MoS₂ thin film before and after the removal of organic molecules. **a**, XRD patterns of MoS₂ thin film before and after removal of organic molecules. **b,c**, Optical images of the MoS₂ thin films at the same location before (b) and after (c) thermal annealing to remove the organic molecules. Scale bars, 10 μ m.

(10) For completeness, in all the transfer characteristics please add the corresponding leakage current.

Response: Thanks for the helpful suggestion. We have now supplemented the leakage current profiles for all transfer curves in the manuscript (Fig. R4-8). These data have also been included in the revised manuscript (Extended Data Fig. 11 and 22).

Figure R4-8 | Leakage current profiles for solution-processable 2D transistors. **a,b**, I_g - V_{gs} leakage current of 20 individual MoS₂ transistors based on 2D nanosheets exfoliated from crystal powders (a) and large-piece single crystal (b). They correspond to the transfer curves in Fig. 2d and 2h of the manuscript. **c,d**, I_g - V_{gs} leakage current of transistors based on n-type semiconducting MoS₂, MoSe₂, and WS₂ thin films (c) and p-type semiconducting MoSe₂, WSe₂, and MoTe₂ thin films (d). They correspond to the transfer curves of Fig. 5b and 5c in the manuscript. **e**, I_d - V_{gs} transfer characteristics and I_g - V_{gs} leakage current of MoS₂ transistors with solution-processable NbS₂ and evaporated Cr/Au electrodes. It corresponds to the transfer curves in Fig. 5f of the manuscript. **f**, I_g - V_{gs} leakage current of MoS₂ transistors with 40-nm-thick solution-processable 2D GaS and conventional 100-nm-thick thermal SiO₂ gate dielectric. It corresponds to the transfer curves in Fig. 5i of the manuscript.

(11) Are the fabricated FETs and memristors stable in air under ambient conditions? Are the memristor curves reproducible after 10, 30 cycles?

Response: Thanks for the good questions. After exposing to air for about one day, the mobility and current on/off ratio of the MoS₂ transistors remain nearly unchanged while the threshold voltage shifts from -40 V to -15 V (Fig. R4-9a). No further change in the electrical

characteristics was observed in the following two weeks, suggesting good stability in air. The pristine transistor behaviors such as the threshold voltage can be stored by simply re-annealing the sample at 150 °C for one hour in nitrogen (Fig. R4-9b). It is likely to suggest the potential adsorption of moisture or other gas molecules in air.

The memristors appear more stable in air because the PVP/TMD superlattices instead of pure TMD film were used. The organic molecules may be the protection layer to increase air stability. For the MoS₂ memristor, we have tested 6 cycles of set and reset process (Fig. R4-9c). And the memristive behavior became unstable for elongated operations. For future studies to achieve more robust memristor performance, the device fabrication procedure may be optimized including the preparation of bottom electrodes with smaller surface roughness and minimum edge spikes following literature reports (*Nat. Commun.* **2022**, 13, 3037).

Figure R4-9 | Stability of MoS₂ transistors and memristors. **a**, I_d - V_{gs} transfer characteristics of pristine MoS₂ transistor and the same device that has been exposed to air for up to 2 weeks. $V_{ds}=1$ V. **b**, I_d - V_{gs} transfer characteristics of pristine MoS₂ transistor and the transistor that has been exposed to air for 2 weeks and then reannealed at 150 °C for 1 hour. $V_{ds}=1$ V. **c**, I-V curves of MoS₂ memristor tested for 6 switching cycles.

(12) In Figure 12 of the supporting Information it seems that the 2D semiconductor is deposited on the whole substrate. Isn't it selectively deposited using photolithography?

Response: Thanks for pointing out the mistake here and the reviewer is correct on this point. The 2D semiconductor is first deposited on the entire substrate which is then patterned by photolithography and etched into the discrete stripes of desired shape (Fig. R4-10). Therefore, we have now updated the schematic illustration in the Extended Data Fig. 19 to show the correct device fabrication process.

Figure R4-10 | Solution-processable integration of 2D metals and 2D semiconductors. **a,b**,

Schematic illustration of the structure (a) and fabrication process (b) of back-gate thin-film transistors consisting of 2D metal and 2D semiconductor thin films.

(13) In Figure 13 of the Supporting Information the authors report the XRD pattern of GaS after annealing at 200 °C, showing the preserved superlattice structure. However, in a previous work of some of the present authors (*Nature*, 562(7726), 254-258, 2018) PVP and ammonium molecules were decomposed from a MoS₂ film after annealing at 200 °C. Were the annealing conditions (ramp rate, used gases,..) the same in this previous work and in the present work? Please clarify this point that might be confusing for the readers.

Response: Thanks for the great suggestion. In brief, the thermal stability differs among these 2D organic/inorganic hybrid superlattice structures. For example, PVP/GaS superlattice structure is more thermodynamically stable than the PVP/MoS₂ structure. The PVP/GaS superlattice can be preserved after annealing at 200 °C and only shows slight interlayer shrinkage after annealing at 300 °C (Fig. R4-11a). By contrast, the PVP/MoS₂ superlattice can be fully converted to pure MoS₂ structure after annealing at 200-300 °C (Fig. R4-11b), as also seen in the previous report (*Nature* **2018**, 562, 254). Among all these exfoliated 2D nanosheets, the temperature for removing organic molecules seems to differ from one to another. This is a very interesting phenomenon and probably related to the interaction force between PVP molecules and the 2D inorganic crystal lattice, which is one of the research topics in our following study.

Accordingly, we have now added descriptions in the figure caption to state the difference in thermal stability between PVP/GaS and PVP/MoS₂ structures (Extended Data Fig. 20).

Figure R4-11 | Thermal stability of the PVP/GaS and PVP/MoS₂ superlattice structures. a,b, XRD patterns of PVP/GaS film (a) and PVP/MoS₂ film (b) after deposition, annealed at 200 °C and 300 °C. The vertical lines indicate the peak position of pristine GaS and MoS₂ crystals.

(14) In some points, the manuscript would benefit from language editing.

Response: Thanks for the suggestion. Accordingly, we have now made thorough edits, corrected grammatical mistakes, and polished the text in the revised manuscript. We hope our revisions have further improved the clarity of the scientific content in the manuscript.

Together, we greatly appreciate all the constructive comments from the reviewer, which have motivated us to finely polish the statements and further strengthen our manuscript. According to those insightful suggestions, we have now carefully revised the manuscript, highlighted all the changes we made, and thus hope to contribute a valuable article to the community.

Reviewer #1 (Remarks to the Author):

Authors addressed my concerns. No further questions from my end.

Reviewer #2 (Remarks to the Author):

The authors have provided solid answers and made additional characterization to address the referees concerns. I therefore support publication of the revised manuscript in Nature comms. However, I would like to suggest the authors a few minor points:

- one of the referees asked for the weight ratio; I am not sure the authors provided the correct answer, which should be given in wt% or %w/w.
- the memristor shows relatively poor performance, eg endurance > 100 cycles should be achieved. As this work does not focus on the optimization of the device, it would be fair for the community to clearly state in the main text that the memristor requires further optimization, which beyond the scope of this work (as clearly stated in the answer to the referees).
- the PL in the Raman can be shifted by changing the excitation wavelength (usually moving towards the UV, eg 488nm or 325 nm)

Please note: I co-reviewed this manuscript with one of the reviewers who provided the listed reports. This is part of the Nature Communications initiative to facilitate training in peer review and to provide appropriate recognition for Early Career Researchers who co-review manuscript

Reviewer #3 (Remarks to the Author):

Reviewer #4 (Remarks to the Author):

The authors made significant efforts to answer in detail to all the raised questions.

I have few more comments:

- 1) I recommend the authors to remove Fig. R4-3 a, which is misleading as the data presented do not refer to transistors prepared by liquid metal electrochemical exfoliation.
- 2) FT-IR spectrum should be performed on MoS₂ after annealing at 300°C to see if PVP and THAB are still present in the channel material of the prepared transistors.
- 3) The authors mention that the intercalation of 100mg MoS₂ powder is about three times faster than that of bulk MoS₂. The authors should add the approximate weight of the bulk MoS₂ piece used in the intercalation experiments for a fair comparison between the exfoliation times.

Response to Reviewer 1:

General comments. *Authors addressed my concerns. No further questions from my end.*

Response: We thank the reviewer for carefully reviewing our manuscript and for the support of the publication on *Nature Communications*.

Response to Reviewer 2 and Reviewer 3 (co-reviewer):

General comments. *The authors have provided solid answers and made additional characterization to address the referees concerns. I therefore support publication of the revised manuscript in Nature comms. However, I would like to suggest the authors a few minor points:*

Please note: I co-reviewed this manuscript with one of the reviewers who provided the listed reports. This is part of the Nature Communications initiative to facilitate training in peer review and to provide appropriate recognition for Early Career Researchers who co-review manuscripts.

(Reviewer 3) I co-reviewed this manuscript with one of the reviewers who provided the listed reports. This is part of the Nature Communications initiative to facilitate training in peer review and to provide appropriate recognition for Early Career Researchers who co-review manuscripts

Response: We thank both reviewers for carefully reviewing the manuscript and for the support of the publication on *Nature Communications*. We especially appreciate the specific questions raised by the reviewer, and welcome the opportunity to address these questions and describe the changes we have made accordingly in the manuscript.

Specific comments.

(1) One of the referees asked for the weight ratio; I am not sure the authors provided the correct answer, which should be given in wt% or %w/w.

Response: Thanks for pointing out the issue. The ratio of powder/liquid metal has now been updated as weight percentage.

On page 14 of the revised manuscript: *“Then the selected 2D powders (e.g., MoS₂, Bi₂Se₃, In₂Se₃, HfS₂, SnS₂, MoO₃, and GaS crystal powders) were added to the liquid metal with a powder weight of 17% w/w (the typical range is 5% to 50%, or powder/liquid metal weight ratio of 1:20 to 1:1).”*

(2) The memristor shows relatively poor performance, eg endurance > 100 cycles should be achieved. As this work does not focus on the optimization of the device, it would be fair for the community to clearly state in the main text that the memristor requires further optimization, which beyond the scope of this work (as clearly stated in the answer to the referees).

Response: Thanks for the comment. Agreed and revised.

On page 12 of the revised manuscript: *“However, the long-term operation stability of the memristors remains a challenge, which requires the further optimization of device*

fabrication process such as bottom electrodes with smaller surface roughness and reduced edge spikes.”

(3) The PL in the Raman can be shifted by changing the excitation wavelength (usually moving towards the UV, eg 488nm or 325 nm)

Response: Thanks for the great suggestion. We agree with the reviewer about shifting PL peak in the Raman spectrum by changing excitation laser wavelength. However, unfortunately, we currently do not have access to a laser source with wavelength shorter than 532 nm. Therefore, the existence of organic molecules in the hybrid superlattice was confirmed by FT-IR spectra instead (refer to the previous response to Reviewer 4).

Response to Reviewer 4:

General comments. *The authors made significant efforts to answer in detail to all the raised questions. I have few more comments:*

Response: We thank the reviewer for carefully reviewing the manuscript and considering that we made “*significant efforts*” to all the questions. We especially appreciate the specific questions raised by the reviewer, and welcome the opportunity to address these questions and describe the changes we have made accordingly in the manuscript.

Specific comments.

(1) I reccomand the authors to remove Fig. R4-3a, which is misleading as the data presented do not refer to transistors prepared by liquid metal electrochemical exfoliation.

Response: Thanks for the suggestion. Agreed and revised.

(2) FT-IR spectrum should be performed on MoS₂ after annealing at 300°C to see if PVP and THAB are still present in the channel material of the prepared transistors.

Response: Thanks for the suggestion. Accordingly, we have now collected the FT-IR spectrum of the MoS₂ thin films after thermal annealing (Fig. R4-1). The removal of PVP and THAB molecules from the MoS₂ thin film can be confirmed by the disappearance of corresponding signature peaks.

Figure R4-1 | Comparison of FT-IR spectra collected from as-deposited MoS₂ thin film and that after thermal annealing at 300 °C.

(3) The authors mention that the intercalation of 100mg MoS₂ powder is about three times faster than that of bulk MoS₂. The authors should add the approximate weight of

the bulk MoS₂ piece used in the intercalation experiments for a fair comparison between the exfoliation times.

Response: Thanks for the suggestion. To clarify, we used a similar weight (~100 mg) for both bulk MoS₂ single crystal and crystal powders to compare the intercalation rate. Accordingly, we have now updated the relevant discussion in the revised manuscript to show these experimental details.

On page 7 of the revised manuscript: “*For example, the powder intercalation completes in ~1 hour (weight ~100 mg) which is about 2-3 times faster than that using large-size bulk single crystal of similar weight (~3 hours).*”